biomathematics/computational biology/systems biology

stochastic gene expression, first-passage times, chemical master equation, stochastic models, fractional killing

**Author for correspondence:**
Jiajun Zhang
e-mail: zhjiajun@mail.sysu.edu.cn

# Stochastic fluctuations in apoptotic threshold of tumour cells can enhance apoptosis and combat fractional killing

## Baohua Qiu[1,2], Tianshou Zhou[1,2] and Jiajun Zhang[1,2]

[1]School of Mathematics, Sun Yat-Sen University, Guangzhou 510275, People's Republic of China
[2]Key Laboratory of Computational Mathematics, Guangzhou, Guangdong Province, People's Republic of China

(iD) JZ, 0000-0001-7107-4814

Fractional killing, which is a significant impediment to successful chemotherapy, is observed even in a population of genetically identical cancer cells exposed to apoptosis-inducing agents. This phenomenon arises not from genetic mutation but from cell-to-cell variation in the activation timing and level of the proteins that regulates apoptosis. To understand the mechanism behind the phenomenon, we formulate complex fractional killing processes as a first-passage time (FPT) problem with a stochastically fluctuating boundary. Analytical calculations are performed for the FPT distribution in a toy model of stochastic p53 gene expression, where the cancer cell is killed only when the p53 expression level crosses an active apoptotic threshold. Counterintuitively, we find that threshold fluctuations can effectively enhance cellular killing by significantly decreasing the mean time that the p53 protein reaches the threshold level for the first time. Moreover, faster fluctuations lead to the killing of more cells. These qualitative results imply that fluctuations in threshold are a non-negligible stochastic source, and can be taken as a strategy for combating fractional killing of cancer cells.

## 1. Introduction

Resistance to chemotherapeutic agents remains a major obstacle to effective cancer treatment. Much effort has been devoted to understanding resistance mechanisms to improve the therapeutic effect. Previous studies considered that drug resistance emerges due to specific mutations in a subset of tumour cells, and it is

**Figure 1.** Threshold crossing can be modelled as a first-passage time (FPT) problem. (*a*) A realistic example involving FPT, where cells must reach a threshold level of *p53* to execute apoptosis and this apoptotic threshold changes with time [3], here only shown its fluctuating extent. (*b*) Two gene models, where the model for gene *p53* assumes that the gene is expressed in a burst manner whereas that for gene *A* assumes that the gene is expressed in a constitutive manner. (*c*) A FPT problem with fluctuating threshold, where *A*(*t*) represents the threshold curve that *p53*(*t*) hits for the first time, and the inset shows the FPT distribution. (*d*) A one-dimensional FPT problem with fluctuating threshold is transformed into a two-dimensional FPT problem with fixed threshold. The shadow region represents an absorbing domain of FPT in the (*p53*, *A*) plane, defined by $\mathcal{D} = \{(p53, A)|p53 \geq A\}$, where an event is triggered once p53 crosses *A*, the line of $A = p53$ represents the boundary of region $\mathcal{D}$, and arrows represent the possible directions of threshold crossing. (*e*) The expression level of gene *p53* changes over time, where the dashed red line represents a critical threshold that the gene product crosses, and the inset shows the FPT distribution. (*f*) Schematic for an absorbing domain of FPT, where the empty circles with arrow represent threshold crossing.

those mutated cells that survive during chemotherapy treatment [1]. However, recent experimental investigations into genetically identical populations of tumour cells exposed to apoptosis-inducing agents revealed that drug resistance also emerges through mechanisms of non-genetic mutations, often through stochastic fluctuations in key factors in response to drugs. Drug resistance means that some cells are killed while others survive during treatment. This phenomenon is known as fractional killing [2].

Single-molecule measurement technologies have shed much light on the underlying molecular mechanisms of cell-to-cell variability in fractional killing [2–5]. For example, experiments verified that genetic mutations in BCR-ABL can give rise to fractional killing of cancer cells or lead to the drug ineffectiveness, but in two-thirds of cases no genetic mutations was found [6]. In the case of apoptosis mediated by tumour necrosis factor-related apoptosis-inducing ligand, it is common that some of the tumour cells of a clonal population are killed while the others survive [2]. In human cell lines, fractional killing arises from cell-to-cell variability in the timing and probability of death, and this variability is thought to originate from the differences in the levels of the proteins that regulate receptor-mediated apoptosis [3]. Another experimental observation is that the cell-to-cell variability in p53 dynamics can result in fractional killing, where a cell's death probability depends on the time and level of p53 and the cell must reach a fluctuating threshold to execute apoptosis (referring to figure 1*a*) [3]. In spite of these case-to-case experimental efforts, how non-genetic variability in the timing and level of key proteins regulating apoptosis impacts fractional killing of cancer cells remains to be fully understood, and model efforts are required to address this intriguing yet important issue, especially in the case of fluctuations in apoptotic threshold.

As is well known, for many cancer types, the p53 transcription factor is a key regulator in the cellular response to DNA damage induced by chemotherapy [7]. Experimental evidence supports that increasing upstream p53 abundance can trigger the transcription of multiple genes in various downstream programmes including cell apoptosis and cell-cycle arrest [8]. Previous works suggested a threshold mechanism where the choice between different programmes depends on p53 protein levels [9,10]. In

the corresponding models, low levels of p53 trigger cell-cycle arrest and high levels of p53 lead to apoptosis. Subsequently, some studies [11,12] showed that the dynamics of p53 plays a role in the specificity of the response with pulsed p53 favouring DNA repair and cell-cycle arrest genes and sustained p53 triggering activation of senescence and apoptotic genes. Recently, Paek *et al.* [3], used live-cell imaging to investigate the role of p53 dynamics in fractional killing of colon cancer cells in response to chemotherapy. They showed that both surviving and dying cells reach similar levels of p53, implying that cell death is not determined by a fixed p53 threshold. Conversely, a cell's death probability depends on the time and levels of p53. They also showed that cells must reach a threshold level of p53 to execute apoptosis and this threshold increases with time. The increase in p53 apoptotic threshold is due to drug-dependent induction of anti-apoptotic genes, predominantly in the inhibitors of apoptosis family. These quantitative experiments call for a corresponding modelling effect that addresses the question of how fluctuations in apoptotic threshold affects fractional killing of cancer cells.

In order to address this issue, we first formulate complex fractional killing processes as a first-passage time (FPT) problem and then analyse a simplified model of stochastic p53 dynamics, where the cancer cell is killed only when the p53 expression level crosses a fluctuating apoptotic threshold. Analytical calculations are performed for the FPT distribution in this model. Counterintuitively, we find that fluctuations in apoptotic threshold can effectively enhance cellular killing by significantly decreasing the mean time that the p53 protein reaches the threshold level for the first time. And faster fluctuations can lead to the killing of more cells. These qualitative results indicate that stochastic fluctuations in apoptotic threshold are a non-negligible noisy source that can facilitate killing of cancer cells. Therefore, tuning this variability would be a potential strategy for combating fractional killing and thus improving drug efficacy.

# 2. Material and methods

## 2.1. Modelling fractional killing processes as a FPT problem

Fractional killing generally results from the cross-talk between complex apoptosis and survival pathways. These complexly structured and heterogeneous processes as well as the paucity of experimental data hamper efforts to construct detail models. However, fractional killing processes are essentially threshold-crossing events. To reveal the essential mechanism of how fluctuating threshold impacts the dynamics of threshold crossing, we consider a toy model of gene regulation (referring to figure 1*b*), where a timing event is triggered once the expression level of a gene (denoted its product by p53) crosses the expression level of another gene (denoted its product by A) for the first time. Indeed, simple mathematical models are important tools towards understanding the essential mechanisms of important biological processes such as cell apoptosis and interpreting experimental phenomena [13–16]. They can also provide guidelines for experimental designs with a growing interest in combining clinical and molecular data.

Specifically, we use a stochastic model of p53 gene expression to investigate the effect of stochastic fluctuations in apoptotic threshold on fractional killing of cancer cells. This model takes explicitly into account 'molecular noise' in the p53 protein that regulates apoptosis in the emergence of drug resistance during treatment, and variability in apoptotic threshold. Indeed, fluctuating threshold has a strong biological background and is ubiquitous in biological regulatory systems. For example, we consider a representative activity function of Hill type [17–19] activation $= Z^n/(Z^n + K^n)$, where $n$ is the Hill coefficient, and $K$ is a function of stochastically generated external signals $S$, and represents fluctuating thresholds of $Z$. For a gene whose expression must reach fluctuating threshold level, expression noise and threshold fluctuations can all lead to variability in the event timing. This raises questions: how these two stochastic origins impact threshold crossing, and which regulatory strategies can control variability in the event timing. Most previous studies have focused on the first-passage properties of stationary threshold crossing [17,18,20–22], while comparatively very few studies have investigated how fluctuating threshold impacts timing precision and mean FPT that reached the threshold for the first time, theoretically [22–26] or experimentally [3].

Now, we formulate the stochastic temporal timing of events as FPT problem to a fluctuating threshold. Denote by $p53$ and $A$ the p53 protein and the apoptotic threshold, respectively. Cells must reach an apoptotic threshold level of $p53$ to execute apoptosis, and this threshold fluctuates with time. Assume that $\{p53(t)\}_{t \geq 0}$ is a temporally homogeneous stochastic process with initial $p53_0$, and $\{A(t)\}_{t \geq 0}$ represents a fluctuating threshold (boundary or barrier) with initial $A_0$. Without loss of generality, we

here set $p53_0 < A_0$. This setting is natural since $A$ represents the critical threshold that $p53$ will cross. Note that the union of two trajectories $p53(t)$ and $A(t)$, $(p53(t), A(t))$ constitutes a new system. Define $T$ as the time that trajectory $p53(t)$ hits trajectory $A(t)$ for the first time, i.e.

$$T = \min\{t : p53(t) \geq A(t) | p53(0) = p53_0, A(0) = A_0\} \tag{2.1}$$

which is called the first-passage time (FPT) [20–22]. Apparently, $T$ is a random variable since both $p53(t)$ and $A(t)$ are stochastic, referring to figure 1$a$. The left issue is how the distribution of $T$ including statistical quantities is correlated to stochastic dynamics of $p53(t)$ and $A(t)$. Our basic idea is to transform a one-dimensional FPT problem with fluctuating threshold into a two-dimensional FPT problem with a fixed boundary.

For analysis convenience, we consider a two-gene expression model to mimic $p53$-induced tumour cell apoptosis, in which $p53$ molecules are produced in a burst manner whereas $A$ molecules are generated in a constitutive manner. The produced counts of protein molecule $A$ are used to construct a stochastically fluctuating threshold that the molecular number of protein $p53$ reaches. Assume that $p53(t) \in \{0, 1, 2, \ldots\}$ is the level of protein $p53$ at time $t$, and protein $p53$ is generated with a Poisson rate $g_{p53}^{(m)}$ (where superscript $(m)$ means that feedback regulation is considered, but it may be omitted in the absence of feedback regulation) and degrades at a constant rate $d_{p53}$. The translation burst approximation is based on the assumption of short-lived mRNAs, meaning that each mRNA degrades instantaneously after producing a burst of $B$ protein molecules, where $B$ follows a geometric distribution [27–30],

$$P_{B=k} \equiv P_B(B = k) = \frac{b^k}{(1+b)^{k+1}}, \quad k = 0, 1, 2, \cdots \tag{2.2}$$

with $b$ mean translation burst size. Thus, $P_{B \geq k} \equiv P_B(B \geq k) = (b/(1+b))^k, \quad k = 0, 1, 2, \cdots$. Similarly, assume that $A(t) \in \{0, 1, 2, \ldots\}$ represents the level of protein $A$ at time $t$, and follows a Poisson distribution with two characteristic parameters $g_A^{(n)}$ (where the meaning of superscript $(n)$ is similar to that of superscript $(m)$) and $d_A$, representing, respectively, the transcription and degradation rates of protein $A$ when $A(t) = n$. Moreover, the time evolution rule of $(p53(t), A(t))$ is defined as follows: $(p53(t), A(t))$ starting from $(p53(t) = m, A(t) = n)$ with $m < n$ at time $t$ is updated through the following probabilities of timing events in the infinitesimal time interval $(t, t + dt]$,

$$\left.\begin{array}{l} P(p53(t+dt) = m+B, A(t+dt) = n | p53(t) = m, A(t) = n) = g_{p53}^{(m)} dt; \\[4pt] P(p53(t+dt) = m-1, A(t+dt) = n | p53(t) = m, A(t) = n) = m d_{p53} dt; \\[4pt] P(p53(t+dt) = m, A(t+dt) = n+1 | p53(t) = m, A(t) = n) = g_A^{(n)} dt; \\[4pt] P(p53(t+dt) = m, A(t+dt) = n-1 | p53(t) = m, A(t) = n) = n d_A dt; \end{array}\right\} \tag{2.3}$$

The apoptosis event occurs if the cumulative number of protein $p53$ molecules exceeds the number of protein $A$ molecules. Here, $A(t)$ described event threshold is not a constant but fluctuates over time. Next, we will focus to investigate the effect of the noise in $A(t)$ on threshold-crossing events, and compare the FPT characters between two cases of fluctuating (i.e. $A(t)$ stochastically changes) and fixed (i.e. $A(t)$ = constant) threshold. Note that the more threshold-crossing events there are, the more cancer cells are killed, and otherwise, the fewer cancer cells are killed.

## 2.2. Master equation for FPT problem with a fluctuating threshold

The FPT problem with fluctuating threshold arises in many scientific fields such as biology, statistics and engineering. However, in contrast to fixed threshold FPT problem, it seems to us that there have been no methods to handle fluctuating threshold FPT problem. For the above example, we successfully transform a one-dimensional FPT problem with a fluctuating threshold into a two-dimensional FPT problem with a fixed boundary. It is worth pointing out that this transform can easily be extended to a more complex case.

Now, we introduce an absorption domain $\mathcal{D}$, which consists of those points $(p53, A)$ that satisfies $p53 \geq A$, that is, $\mathcal{D} = \{(p53, A) | p53 \geq A\}$. Let $P_{m,n}$ represent the probability that a two-dimensional system is at state $(m, n)$ at time $t$, i.e.

$$P_{m,n}(t) = \text{Prob}\{p53(t) = m, A(t) = n | p53(0), A(0)\} \tag{2.4}$$

$P_{m,n}(t)$ is sometimes also denoted by $P_S(t)$, i.e. $P_S(t) = P_{m,n}(t)$, where $S = (p53, A)$ represents state. Note that the survival probability is equal to the sum of the probabilities of all the states that do not belong to the absorbing region, i.e. $\mathbb{S} = \sum_{S \notin \mathcal{D}} P_S(t)$, and the probability density function for the FPT (denoted by $f_T(t)$) satisfies $f_T(t) = \text{Prob}\{T \leq t\}$.

The relation between the protein molecules $p53$ and $A$ can be considered as a trajectory in the domain $\{(p53, A)|p53(t) < A(t)\}$. The corresponding forward master equation (FME) describing the time evolution of protein pair $p53$ and $A$ can be described as the following master equation [20,29,30]:

$$\frac{\mathrm{d}P_{m,n}(t)}{\mathrm{d}t} = \sum_{i=0}^{m-1} g_{p53}^{(i)} P_{B=m-i} P_{i,n}(t) + g_A P_{m,n-1}(t) + (m+1)d_{p53} P_{m+1,n}(t)$$
$$+ (n+1)d_A P_{m,n+1}(t) - (g_{p53}^{(m)} P_{B\geq1} + g_A + md_{p53} + nd_A)P_{m,n}(t), \tag{2.5}$$

where $m < n$. Thus, FPT distribution ($f_T(t)$) can be formally expressed as [22,31,32]

$$f_T(t) = \sum_{m\geq0} (m+1)d_A P_{m,m+1}(t) + \sum_{n>m\geq0} g_{p53}^{(m)} P_{B\geq n-m} P_{m,n}(t). \tag{2.6}$$

In numerical simulation, we constrain $n = 1, 2, \ldots, C$ (where $C$ is a pre-given positive integer), implying that $m = 0, 1, 2, \cdots, C - 1$.

## 2.3. Statistical quantities of FPT distribution

Although the FPT distribution in principle provides complete characterization of the threshold-crossing event timing, we are particularly interested in the lower-order statistical moments of FPT distribution ($f_T(t)$). Starting from a general FME, we can obtain analytical formulae for the first- and second-order moments of FPT. For this, we first establish the relation between distribution $f_T(t)$ and state $S$, and then give the formal expression of $f_T(t)$. Assume that all states $\{S(t)\}_{t\geq0}$ with $S(t) = (p53(t), A(t))$ constitute a Markov process. The vector form of FME can be then written as [20,29,30]

$$\frac{\partial \mathbf{P}(t)}{\partial t} = \mathbf{M}\mathbf{P}(t), \tag{2.7}$$

where $\mathbf{P}(t)$ is a column vector consisting of all $P_S(t)$, and $\mathbf{M}$ is a certain linear operator, depending on a process of interest. Here, every component of $\mathbf{P}(t)$ is the probability that the system $\{S(t)\}_{t\geq0}$ arrives at the absorbing domain $\mathcal{D}$ at time $t$ and $\mathbf{M}$ is actually a state transition matrix. $P(S_f, t|S, t_0)$ is denoted by the probability that the state $S(t)$ reaches the absorbing state $S_f$ at time $t$, given an initial state $S = S(t_0)$ at time $t_0$ with $S(t_0) = (p53_0, A_0)$ ($t_0 = 0$ can be set). Let $\mathbb{S}(t, S_f|S, t_0)$ be the survival probability that the trajectory $\{S(t)\}$ starting from $S$ at time $t_0$ has not yet been absorbed to state $S_f$ at time $t$, that is, $\mathbb{S}(t, S_f|S, t_0) = \sum_{S' \neq S_f} P(S', t|S, t_0)$. By the definition of survival probability, we have $\text{Prob}\{T \leq t\} = 1 - \mathbb{S}(t, S_f|S, t_0)$. Thus, the probability density function of the FPT, $f_T(t)$, is given by (see electronic supplementary material, S1 for more details)

$$f_T(t) = \mathbf{W}^T\mathbf{P}(t) = \mathbf{W}^T\exp(\mathbf{M}t)\mathbf{P}(0), \tag{2.8}$$

where $\mathbf{W}$ is a column vector of transition rates from all accessible states to the absorbing state and the superscript T represents transpose [22,31,32].

Once $f_T(t)$ is given or found, raw moments of random variable $T$ are given by (see electronic supplementary material, S1 for derivation)

$$\langle T^k \rangle = \int_0^{+\infty} t^k f_T(t)\,\mathrm{d}t = k!(-1)^k \mathbf{e}^T (\mathbf{M}^{-1})^k \mathbf{P}(0), \quad k = 1, 2, \cdots \tag{2.9}$$

with $\mathbf{e}^T = [1, 1, \cdots, 1]$ constant vector. This indicates that the moments of FPT can be calculated directly based on the FME once initial transition probability $\mathbf{P}(0)$ is set. In particular, the *timing mean*, i.e. mean FPT (MFPT), is obtained by

$$\text{MFPT} = \langle T \rangle = -\mathbf{e}^T\mathbf{M}^{-1}\mathbf{P}(0), \tag{2.10}$$

which is a statistical quantify of our main interest. Moreover, the intensity of the noise in $T$ (defined as the ratio of variance over the square of mean), which represents the *timing variability* or reflects the precision

in the event timing, is calculated by

$$CV_T = \frac{\langle T^2 \rangle - \langle T \rangle^2}{\langle T \rangle^2} = \frac{2\mathbf{e}^T (\mathbf{M}^{-1})^2 \mathbf{P}(0)}{[\mathbf{e}^T \mathbf{M}^{-1} \mathbf{P}(0)]^2} - 1. \tag{2.11}$$

Other higher-order moments such as skewness and kurtosis can be also formally given, detailed in electronic supplementary material, S1. Obviously, the key to calculating these statistical quantities is to calculate the inverse of matrix $\mathbf{M}$. Note that the more the mean FPT $\langle T \rangle$ is, the fewer cancer cells are killed, and the smaller the timing variability $CV_T$ is, the more precise threshold crossing is.

It should be pointed out that the molecule number of protein $p53$ or $A$ may be infinite in theory, implying that $\mathbf{M}$ in equation (2.7) is an infinite-dimensional matrix. Therefore, equations (2.9)–(2.11) have only theoretical significance since they give only the formal expressions of FPT distribution and statistical quantities. Owing to such infinity, the FPT problem we study here is essentially different from a traditional FPT problem in which matrix $\mathbf{M}$ is finitely dimensional due to the fixed threshold. The infinite-dimensional FPT problem is in general intractable, thus it is needed to develop computational methods. Here, we propose a so-called truncation approach to solve this tough problem. This approach is developed based on the finite state projection [33].

## 2.4. An efficient method for solving FPT problem with a fluctuating threshold

For above FPT model, we introduce our truncation method to solve FPT problem, which can be generic and applied to more complex cases. Next algorithm description is given.

First, the finite state projection approach [33] tells us that matrix $\mathbf{M}$ can be replaced by a $k \times k$ submatrix $\mathbf{M}_k$, so that the approximation $\mathbf{P}(t) \approx \exp(\mathbf{M}_k t)\tilde{\mathbf{P}}(0)$ holds, where $\tilde{\mathbf{P}}(0)$ replaces the original $\mathbf{P}(0)$ in some order. As a result, the state vector $(p53_i, A_i)$ constitutes a finite state projection, where $i \in \{1, 2, \ldots, k\}$.

Second, we define $\Gamma_k = \mathbf{e}^T \exp(\mathbf{M}_k t)\tilde{\mathbf{P}}(0)$, which represents the sum of the components of vector $\mathbf{P}(t)$. According to the finite state projection approach, we can prove that if $\Gamma_k \geq 1 - \varepsilon$ with $\varepsilon$ being a small positive number, we have

$$\exp(\mathbf{M}_k t)\tilde{\mathbf{P}}(0) \leq \mathbf{P}(t) \leq \exp(\mathbf{M}_k t)\tilde{\mathbf{P}}(0) + \varepsilon \cdot \mathbf{e}. \tag{2.12}$$

Based on the above analysis, we develop the following truncation algorithm:

**Inputs**. Propensity functions and stoichiometry for all reactions.
Initial probability density vector $\tilde{\mathbf{P}}(0)$
Final time of interest, $t_f$.
Total amount of acceptable error, $\varepsilon$.
Initial finite set of states, $(p53_0, A_0)$.
Initialize a counter, $k = 0$.
**Step 0** Calculate $\mathbf{M}_k = \text{Submatrix}(\mathbf{M})$, which depends on $(p53_k, A_k)$, and $\Gamma_k = \mathbf{e}^T \exp(\mathbf{M}_k t)\tilde{\mathbf{P}}(0)$.
**Step 1** If $\Gamma_k \geq 1 - \varepsilon$, **Stop**.
$\exp(\mathbf{M}_k t)\tilde{\mathbf{P}}(0)$ approximates the probability $\mathbf{P}(p53_k, A_k, t_f)$ with error $\varepsilon$.
**Step 2** Add more states, $(p53_{k+1}, A_{k+1}) = \text{expand}(p53_k, A_k)$, and take $k \leftarrow k + 1$. Increment $k$ and return to **Step 1**.

Owing to the effectiveness of the truncation approach proposed above (referring to figure 2), we may assume that matrix $\mathbf{M}$ is finitely dimensional (otherwise, we use finitely dimensional matrix, $\mathbf{M}_k$). To derive the expression of matrix $\mathbf{M}$ in the above gene model, a special absorbing domain is considered by

$$\mathcal{D}_1 = \{(p53, A) | p53 = p53(t) \geq A(t) = A\}. \tag{2.13}$$

Electronic supplementary material, S1 performs analysis for three other kinds of absorbing domains.

Introduce the numerical cut-offs for the numbers of proteins $p53$ and $A$, respectively: $p53_{max}$ for $p53(t)$ and $A_{max}$ for $A(t)$, and without loss of generality, assume $p53_{max} = A_{max} = C$ (a known integer). Therefore, as for the above gene model (figure 1$f$), we have $m \in \{0, 1, 2, 3, \ldots, C-1\}$ and $n \in \{1, 2, 3, \ldots, C\}$. That means that the corresponding finite state-space for birth–death process can be considered by

$$\Omega = \{(m,n) | m < n, \quad m = 0, 1, 2, \cdots, C-1, n = 1, 2, \cdots, C\}. \tag{2.14}$$

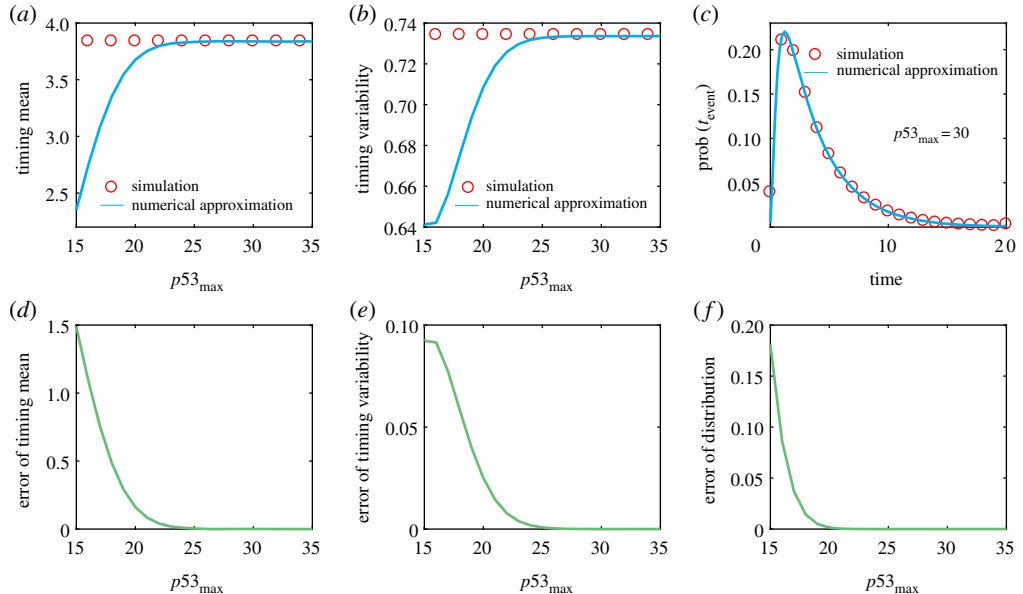

**Figure 2.** Verification of the effectiveness of the truncation algorithm with the model described in figure 1b. (a) Timing mean as a function of $p53_{max}$. (b) Timing variability as a function of $p53_{max}$. (c) The FPT distribution for $p53_{max} = 30$. (d) The difference between exact and approximate results of timing mean as a function of $p53_{max}$. (e) The difference between exact and approximate results of timing variability as a function of $p53_{max}$. (f) The Kullback–Leibler divergence between exact and approximate FPT distribution as a function of $p53_{max}$. In (a–f), empty circles represent the results obtained by Gillespie stochastic simulation and are therefore viewed as 'exact', whereas the curve represents the results obtained by our algorithm and are therefore viewed as 'approximate'. The parameter values are set as $g_{p53} = 5$, $d_{p53} = 1$, $b = 1$, $g_A = 10$, $d_A = 1$. And the mean value of $A$ (i.e. event threshold) is $A_{threshold} = 10$ in the $p53$ cut-offs, $p53_{max} = 15 \sim 35$.

By this finite state-space, vector $\mathbf{P}(t)$ is rewritten as $\mathbf{P} = [\mathbf{P}_{*,1}, \mathbf{P}_{*,2}, \mathbf{P}_{*,3}, \ldots, \mathbf{P}_{*,C}]^{\mathrm{T}}$, where $\mathbf{P}_{*,k}$ represents $\mathbf{P}_{*,k} = [P_{0,k}, P_{1,k}, \ldots, P_{k-1,k}, 0, \ldots, 0]$, $k \in \{1, 2, \ldots, C\}$, and time $t$ is omitted. Also, we introduce an operator, denoted by $\mathbb{L}^{(i)}$ ($i = 1, 2, \cdots, n$), which acts on matrix with the operation rule being: $\mathbb{L}^{(i)}\mathbf{M} = \mathbf{M}^{(i)}$, where $\mathbf{M}^{(i)}$ is a matrix whose order is the same as that of $\mathbf{M}$ but some components are possibly zero, e.g. if $\mathbf{M} = (a_{ij})_{3\times3}$, then $\mathbb{L}^{(2)}\mathbf{M} = \mathbf{M}^{(2)} = (b_{ij})_{3\times3}$, where $(b_{ij})_{2\times2} = (a_{ij})_{2\times2}$ and the other elements are equal to zero. Thus, above matrix $\mathbf{M}$ in equation (2.7) can be expressed as the following form:

$$\mathbf{M} = \begin{bmatrix} \mathbf{D}_1 & \mathbf{U}_1 & & & & \\ \mathbf{L}_1 & \mathbf{D}_2 & \mathbf{U}_2 & & & \\ & \mathbf{L}_2 & \mathbf{D}_3 & \mathbf{U}_3 & & \\ & & \ddots & \ddots & \ddots & \\ & & & \mathbf{L}_{C-2} & \mathbf{D}_{C-1} & \mathbf{U}_{C-1} \\ & & & & \mathbf{L}_{C-1} & \mathbf{D}_C \end{bmatrix}, \tag{2.15}$$

where

$$\begin{cases} \mathbf{U}_i = (i+1)\mathbb{L}^{(i)}(d_A \mathbf{I}_C), & i = 1, 2, \cdots, C-1; \\ \mathbf{D}_i = \mathbb{L}^{(i)}(\mathbf{M}_{g_{p53}} + \mathbf{M}_{d_{p53}} - (g_A + i d_A)\mathbf{I}_C), & i = 1, 2, \cdots, C; \\ \mathbf{L}_i = \mathbb{L}^{(i)}(g_A \mathbf{I}_C)(i = 1, 2, \ldots, C-1), & i = 1, 2, \ldots, C-1. \end{cases}$$

Here, $\mathbf{I}_C$ is an identity matrix. Matrix $\mathbf{M}_{d_{p53}} = d_{p53}\mathrm{diag}([1, 2, \ldots, C-1], 1) - d_{p53}\mathrm{diag}([0, 1 \ldots, C-1], 0)$, where symbol $\mathrm{diag}(\mathbf{v}, k)$ represents that the elements of vector $\mathbf{v}$ are placed on the $k$th diagonal. Note that $k = 0$ corresponds to the main diagonal, $k > 0$ corresponds to above the main diagonal, and $k < 0$ corresponds to below the main diagonal. Matrix $\mathbf{M}_{g_{p53}} = g_{p53}\mathbf{M}_{burst}$, where $\mathbf{M}_{burst} = -P_{B\geq1}\mathbf{I}_C + \sum_{k=1}^{C-1} P_{B=k}\mathrm{diag}(\mathbf{e}_{C-k}^{\mathrm{T}}, -k)$. In the presence of feedback, implying that $g_{p53}$ depends on the molecule number ($m$) of protein $p53$, we have $\mathbf{M}_{g^{(m)}_{p53}} = \mathbf{M}_{burst}\mathbf{G}$, where $\mathbf{G} = \mathrm{diag}([g^{(0)}_{p53}, g^{(1)}_{p53} \ldots, g^{(C-1)}_{p53}], 0)$ (See electronic supplementary material, S1 for the formal expressions of these matrices).

Moreover, given a numerical cut-off $(C)$, the FPT distribution in equation (2.8) can be rewritten as (see electronic supplementary material, S1 for more details)

$$
\begin{aligned}
f_T(t) &= \sum_{m=0}^{C-1}(m+1)d_A P_{m,m+1}(t) + \sum_{n=1}^{C}\sum_{m=0}^{n-1} g_{p53}^{(m)} P_{B\geq n-m} P_{m,n}(t) \\
&= \mathbf{W}_{d_A}^T \mathbf{P}_{m,n}(t) + \mathbf{W}_{g_{p53}}^T \mathbf{P}_{m,n}(t) \\
&\equiv \mathbf{W}^T \mathbf{P}_{m,n}(t),
\end{aligned}
\tag{2.16}
$$

where $\mathbf{W} = [\mathbf{W}_1^T, \mathbf{W}_2^T, \ldots, \mathbf{W}_C^T]^T$ with $\mathbf{W}_n = nd_y \mathbf{1}_n + \sum_{m=0}^{n-1} g_{p53}^{(m)} P_{B\geq n-m} \mathbf{1}_{m+1}$, $n=1,2\ldots,C$. Here, we define a column vector of length $C$, $\mathbf{1}_i = (0,\ldots 0,1,0,\ldots,0)^T$ in which only the $i$th element is equal to 1 and other elements are all zero.

For a given $\mathbf{P}(0)$, mean FPT $\langle T \rangle$ and timing variability $CV_T$ can be calculated by equations (2.10) and (2.11), respectively, where a key step is to calculate the inverse of matrix $\mathbf{M}$ through equation (2.15), while FPT distribution $f_T(t)$ is easily calculated through equation (2.16). In a word, through the calculation of these quantities, we can analyse characteristics of timing events with fluctuating thresholds, including the mean first passage time and variability in the timing. For the sake of simplicity, we will not consider feedback regulation implying that $g_{p53}^{(m)} \equiv g_{P53}$ and $g_A^{(n)} \equiv g_A$ are independent of $m$ and $n$.

# 3. Results

## 3.1. The effectiveness of truncation algorithm

In order to verify the effectiveness of the truncation algorithm proposed above, we perform numerical calculation using the gene model described in figure 1b. Numerical results are shown in figure 2.

Figure 2a,b shows the dependence of mean FPT and timing variability on the cut-off of the protein p53 molecular numbers, respectively. With increasing the cut-off constant, both can well approximate the 'exact' values (empty circles) obtained by the Gillespie stochastic algorithm [34]. For example, the approximate mean FPT is nearly equal to the exact mean FPT at $p53_{max} \approx 23$ whereas the approximate timing variability is nearly equal to the timing variability at $p53_{max} \approx 25$. Figure 2c shows the FPT distribution for a given cut-off constant, $p53_{max} = 30$.

Figure 2d demonstrates the dependence of the difference between the exact mean FPT obtained by the Gillespie stochastic algorithm [34] and the approximate mean FPT obtained by the finite state projection on the cut-off constant $p53_{max}$. We observe that this difference quickly tends to zero as the cut-off constant is beyond some value. The similar change tendency holds for timing variability, referring to figure 2e. In addition, figure 2f more clearly displays that two kinds of FPT distributions are in agreement since the Kullback–Leibler divergence [35] between them tends to zero as the cut-off constant $p53_{max}$ increases, further verifying the effectiveness of the proposed truncation algorithm.

In the remainder of this article, we will use the above numerical method to compute two statistical quantities of FPT distribution, *timing mean* (mean FPT) and *timing variability*. The former characterizes the response time that p53 reaches the apoptosis threshold, i.e. the shorter the *timing mean* (mean FPT) is, the more cells are killed. The latter quantifies the timing precision in threshold crossing. Lower variability implies more robust cell-killing strategy.

## 3.2. Stochastic fluctuations in apoptotic threshold can accelerate tumour cell apoptosis

First, the conception of event threshold is introduced for convenience, which means the average level of protein $A$ that is equal to the ratio of the generation rate $(g_A)$ over the degradation rate $(d_A)$, e.g. the event threshold $A_{threshold} = g_A/d_A$. An event threshold is nothing but a fixed threshold in the deterministic case (i.e. no fluctuations case), but, in the case of fluctuations, an event threshold may not be equal to the fixed threshold, due to the stochastic fluctuations effect.

To show how fluctuating threshold impacts the timing of events, we plot figure 3 where numerical results in the case of fixed threshold are also shown for comparison. Figure 3a shows two curves for the dependence of mean FPT on event threshold in two cases of fixed and fluctuating threshold. And the mean FPT curve of fluctuating threshold case is always below that in the case of fixed threshold as event threshold increases, which implies that threshold fluctuations always shorten the time that regulatory proteins reach a critical threshold, or accelerate the response of intracellular events to

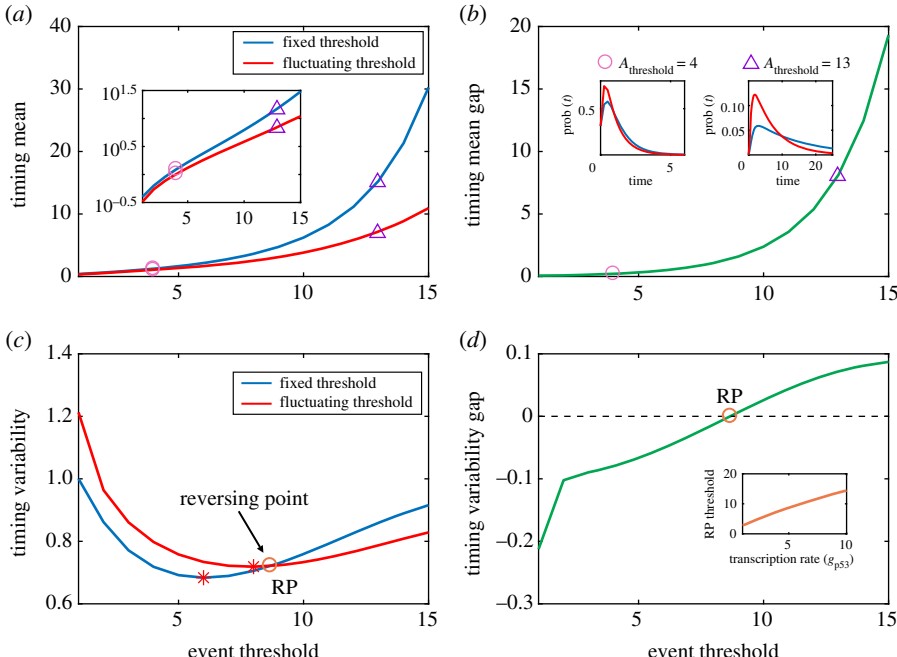

**Figure 3.** Comparison between the effects of fixed and fluctuating thresholds on timing. (*a*) Timing mean as a function of event threshold in two different kinds of fixed and fluctuating threshold, where the inset shows the timing mean as a function of event threshold on the logarithmic scale. (*b*) A different demonstration of the results in (*a*), showing the difference of timing mean in the case of fixed threshold minus that in the case of fluctuating threshold, where two insets show FPT distributions for two different event thresholds (indicated by empty circle and triangle) corresponding to $A_{\text{threshold}} = 4$ and $A_{\text{threshold}} = 13$. (*c*) Timing variability as a function of event threshold in two different kinds of thresholds, where the empty circle is the crossing point of two curves, and stars represent the event threshold that makes timing variability reach the minimum. (*d*) As a supplement of (*c*), the difference of timing variability in the case of fixed threshold minus that in the case of fluctuating threshold, where the inset shows the critical threshold (RP) as a function of transcription rate $(g_{\text{p53}})$. In (*a,c*), the parameter values are set as $g_{\text{p53}} = 5$, $d_{\text{p53}} = 1$, $b = 1$, $p53_{\text{max}} = 30$, $d_A = 1$ and $g_A = 1 \sim 15$. Here, we always keep $d_A = 1$, thus the event threshold is decided by $g_A$, i.e. event threshold $= g_A = 1 \sim 15$. That is, if the fixed threshold is $A_{\text{threshold}} = 10$, then the fluctuating threshold corresponds to $g_A = 10$ and $d_A = 1$. The inset in (*d*) corresponds to $d_{\text{p53}} = 1$, $b = 1$, $A_{\text{threshold}} = g_A = 1 \sim 10$, $d_A = 1$, $g_{\text{p53}} = 1 \sim 10$ and $p53_{\text{max}} = 30$.

external cues. Figure 3*b* further demonstrates the dependence of the difference between their mean FPTs in two cases on event threshold, and this difference monotonically increases with event threshold increased. This difference can be also explained by FPT distribution, such as two special FPT distributions for $A_{\text{threshold}} = 4$ and $A_{\text{threshold}} = 13$, referring to the inset of figure 3*b*, which corresponds to empty circle and triangle indicated, respectively.

Figure 3*c* demonstrates how event threshold impacts variability in the timing. From this figure, we observe that there is a critical event threshold (denoted by RP) such that the timing variability in the case of fluctuating threshold is smaller than that in the case of fixed threshold as the event threshold is beyond RP, but the former is larger than the latter as the event threshold is below RP. In other words, for a high event threshold, threshold fluctuations can reduce the timing variability or can raise the precision in the timing. There is also an even threshold such that timing variability is least (referring to the stars indicated), implying event threshold can make the timing precision reach optimality in both cases of fixed and fluctuating threshold. Figure 3*d*, which is a different demonstration of the results in figure 3*c*, further shows the difference between their timing variabilities in two cases of fixed and fluctuating threshold, which is a monotonically increasing function of event threshold. The dependence of the critical event threshold (RP) on transcription rate, $g_{\text{p53}}$, is shown by the inset of figure 3*d*, demonstrating that the critical event threshold (RP) increases with $g_{\text{p53}}$ increased.

In short, figure 3*a,c* shows our main results, that is, fluctuations in threshold can accelerate the response of intracellular events to external cues by shortening the time that regulatory proteins reach the apoptotic threshold for the first time; fluctuations in high event threshold can raise the timing

precision by reducing timing variability; and there is an even threshold such that the timing variability reaches optimality in both cases of fixed and fluctuating thresholds. These results imply that threshold fluctuations are an important factor affecting the timing of events, and that fluctuations in apoptotic threshold can facilitate the killing of cancer cells.

Note that our above results can also be explained theoretically [36,37]. For the case of fixed threshold, we define $T_{\text{fixed}} = \min\{t: p53(t) \geq \langle A(t) \rangle | p53(0) = p53_0, A(0) = A_0\}$ is the apoptotic timing of random variable $T_{\text{fixed}}$, where $\langle A(t) \rangle$ is mean apoptotic threshold. Obviously, it is different with the defined variable $T$ of equation (2.1) in the case of fluctuating threshold, rewritten by $T_{\text{fluctuating}} = \min\{t: p53(t) \geq A(t) | p53(0) = p53_0, A(0) = A_0\}$. As for nonlinear function, these two mean FPTs are not equal, i.e. $\langle T_{\text{fluctuating}}(A(t)) \rangle \neq \langle T_{\text{fixed}}(\langle A(t) \rangle) \rangle$, implying that the average apoptotic timing response probability is generally not the same as the timing response probability for the average apoptotic threshold signal. Although it is difficult to deduce their size relationship directly theoretically in our model, according to numerical calculation results, we can show this result that the mean FPF in the case of fluctuating threshold is less than that in the case of fixed threshold.

The above model only discusses one case of absorbing domain, referring to figure 1d, but in electronic supplementary material, S1 the other three cases of different absorbing domains are given, and numerical results for their mean FPT and timing variability are also shown, which are a function of event threshold. From electronic supplementary material, figures S6A, S7A and S8A, we can observe that the absorbing domains specified above are smaller than the one in figure 1d. Comparing with the results of figure 3, electronic supplementary material, figures S6–S8 also demonstrate that threshold fluctuations can affect mean FPT, and event threshold can impact timing variability. For a high event threshold, fluctuations in threshold can improve the event response and shorten mean FPT, and the corresponding variability tendency in the timing can raise the timing precision.

## 3.3. Fast fluctuations in apoptotic threshold can lead to killing of more cancer cells

In the above subsection, we have shown that fluctuation in threshold has important influences on event timing. However, factors leading to such fluctuations may be diverse. Here, we focus on investigating the effects of timescales on timing mean and timing variability.

To better understand the characterization of timescale, we first give the definition of timescale in our model. If the production rate ($g_{p53}$ or $g_A$) and degradation rate ($d_{p53}$ or $d_A$) of protein $p53$ or $A$ are simultaneously enlarged by $\alpha_{p53}$ or $\alpha_A$ times, then the factor $\alpha_{p53}$ or $\alpha_A$ is defined as the timescale of protein $p53$ or $A$. In general, the larger the factor $\alpha_{p53}$ or $\alpha_A$ is, the larger are the fluctuations in protein $p53$ or $A$. Therefore, $\alpha_{p53}$ or $\alpha_A$ is an important factor leading to fluctuations in protein $p53$ or $A$. Small $\alpha_{p53}$ or $\alpha_A$ corresponds to slow fluctuations whereas large $\alpha_{p53}$ or $\alpha_A$ to fast fluctuations. We can prove that the variability in event timing depends only on the ratio of $\alpha_A$ over $\alpha_{p53}$, ($\alpha_A/\alpha_{p53}$), independent of their sizes (see electronic supplementary material, S1 for details).

Next, we investigate the influence of timescales on mean FPT and variability in the event timing. Numerical results are shown in figure 4. Specifically, figure 4a,d demonstrates how two timescales of proteins $p53$ and $A$ together affect the mean FPT and timing variability, respectively. We observe that the mean FPT in the case of small $\alpha_{p53}$ is always larger than that in the case of large $\alpha_{p53}$, independent of $\alpha_A$ (referring to the red dashed line). For a fixed yet small $\alpha_{p53}$ (referring to the blue dashed line), the mean FPT in the case of small $\alpha_A$ is also larger than that in the case of large $\alpha_A$. These imply that two kinds of timescales (internal for $p53$ and external for $A$) can all significantly impact the mean FPT. Moreover, a larger external timescale leads to a less mean FPT for small internal timescale, but a smaller internal timescale leads to a larger mean FPT for small external timescale. However, this relationship is different in the case of timing variability, referring to figure 4d. We observe that there is a strip region (indicated by orange) in the plane of $\alpha_{p53}$ and $\alpha_A$, such that the variability in the timing is largest. More precisely, if two boundary lines of this region are denoted as $\ell_1$ and $\ell_2$, which are described by $\alpha_A = a_1\alpha_{p53}$ and $\alpha_A = a_2\alpha_{p53}$, where $a_1$ and $a_2$ are both positive constants satisfying $a_1 < a_2$, the timing variability below $\ell_1$ or beyond $\ell_2$ is less than that in the strip region, and the timing variability below $\ell_1$ is less than that beyond $\ell_2$ (referring to the dashed line with arrow). These indicate that in the outside of the strip region, if the timescale of protein $A$ is dominant, the timing variability becomes smaller, and conversely, if the timescale of protein $p53$ is dominant, the timing variability also becomes smaller. Since timescale factors $\alpha_{p53}$ and $\alpha_A$ determine the noise in proteins $p53$ and $A$ (called intrinsic and extrinsic noise), respectively, both intrinsic and extrinsic noise can significantly contribute to the timing variability, but this contribution depends on

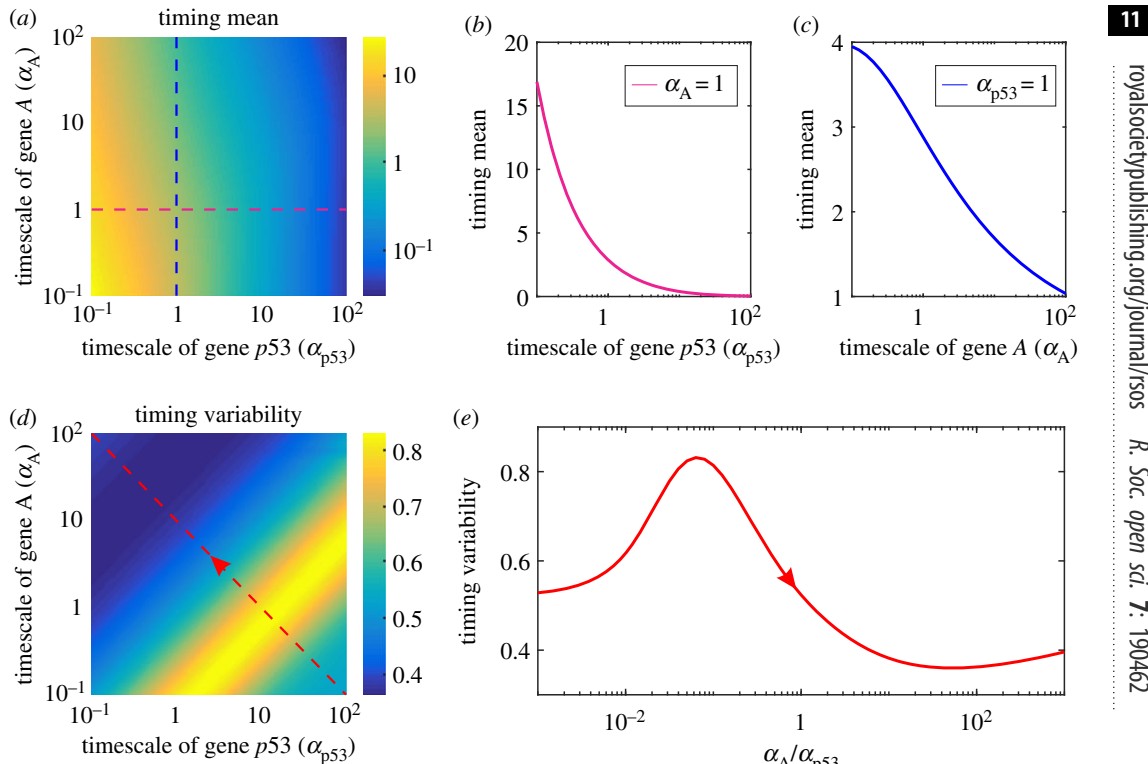

**Figure 4.** Effects of timescales on the timing of events, where $\alpha_{p53}$ and $\alpha_A$ are the timescale of proteins $p53$ and $A$, respectively. (a) Heatmap showing timing mean as a function of $\alpha_{p53}$ and $\alpha_A$. (b) Timing mean as a function of $\alpha_{p53}$, when $\alpha_A = 1$. (c) Timing mean as a function of $\alpha_A$, when $\alpha_{p53} = 1$. (d) Heatmap showing timing variability as a function of $\alpha_{p53}$ and $\alpha_A$. (e) Timing variability as a function of the rate of the timescale of protein $A$ over that of protein $p53$, $\alpha_A/\alpha_{p53}$, where arrow is in agreement with that in (d). In (a–e), parameter values are $g_{p53} = 15$, $d_{p53} = 1$, $b = 1$, $g_A = 20$, $d_A = 1$ and $p53_{max} = 50$. The fluctuating event threshold is $A_{threshold} = 20$.

which noise is dominant. This is an interesting phenomenon similar to the resonance that takes place as the internal frequency is approximately equal to the external frequency [38].

From figure 4b,c, we also observe that mean FPT is a monotonically decreasing function of the timescale of proteins $A$ and $p53$, respectively, and that the former is a convex-downward curve for a timescale $\alpha_{p53}$ when $\alpha_A = 1$, whereas the latter, given $\alpha_{p53} = 1$, is fundamentally a line for a timescale $\alpha_A$. These results are practically special results shown in figure 4a, where $\alpha_{p53} = 1$ and $\alpha_A = 1$, respectively, correspond to blue and red dashed lines. Figure 4b,c shows that internal and external (or threshold) timescales can all shorten mean FPT, further implying that timescales can speed up timing response.

Figure 4e, which corresponds to the red dashed line with arrow in figure 4d, shows how the rate between the timescales of proteins $p53$ and $A$, $\gamma = \alpha_A/\alpha_{p53}$, impacts the variability in the event timing. Interestingly, we observe that there is an optimal rate between external and internal timescales, $\gamma_{critical}$, such as the timing variability is maximal, implying that the precision in the event timing is worst for this optimal timescale. Furthermore, timing variability increases as the rate $\gamma$ satisfying $\gamma < \gamma_{critical}$ increases, whereas it fundamentally decreases when the rate $\gamma$ satisfying $\gamma > \gamma_{critical}$ increases. The former implies that when the rate of external timescale over internal timescale, $\gamma$, is less than the critical rate, $\gamma_{critical}$, this rate weakens the precision in the event timing, and conversely, it fundamentally enhances this precision.

In a word, the timescales of proteins $p53$ and $A$ are two non-negligible factors in the event timing, since they can significantly affect timing precision and mean FPT. And there is a strip region in the plane of external and internal timescales such that the timing variability is largest (implying that the timing precision is worst).

## 3.4. Effects of p53 transcription and degradation rates on fractional killing of cancer cells

In our model, apart from parameters associated with promoter kinetics, there are the protein transcription and degradation rates involved. The curves shown in figure 4 correspond to a special

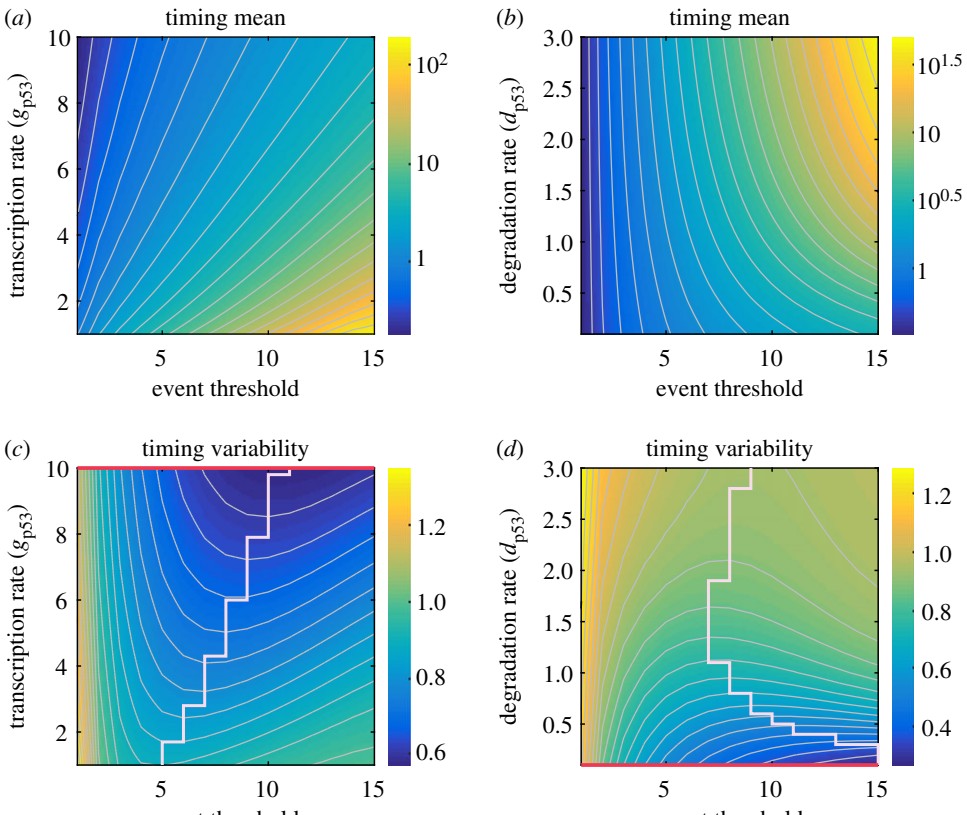

**Figure 5.** Influence of transcription or degradation rate on the timing of events. (*a,c*) Heatmap, respectively, showing timing mean and timing variability as a function of both event threshold and transcription rate ($g_{p53}$). (*b,d*) Heatmap, respectively, showing timing mean and timing variability as a function of event threshold and degradation rate ($d_{p53}$). (*c*) The stair-like line consists of the points corresponding to the star of figure 3*c*, which is an event threshold making timing variability minimum when a special value of $g_{p53}$ is given. (*d*) The meaning of the stair-like line is similar with that in (*c*). The red line represents that for a given event threshold, timing variability reaches the minimum when transcription rate in (*c*) is sufficiently large or degradation rate in (*d*) is smaller. In (*a–d*), the white curves are contour lines, and the parameter values are set as $b = 1$, $d_A = 1$, $p53_{max} = 30$ and $A_{threshold} = 1 \sim 15$ decided by $g_A = 1 \sim 15$. In (*a,c*), we set $g_{p53} = 1 \sim 10$ and $d_{p53} = 1$, whereas in (*b,d*), $g_{p53} = 5$ and $d_{p53} = 0.1 \sim 3$.

value of the transcription or degradation rate of protein. However, these two rates can be regulated by external signals, leading to changes in biologically reasonable intervals. This raises a question: how the two parameters impact the mean FPT and variability in the event timing. Here, we numerically analyse this impact, with results shown in figure 5.

Figure 5*a* (or figure 5*b*) shows a heatmap for the dependence of mean FPT on both event threshold and transcription rate (or degradation rate). For a given special event threshold, mean FPT monotonically decreases, as transcription rate ($g_{p53}$) increases (referring to figure 5*a*) or degradation rate ($d_{p53}$) decreases (referring to figure 5*b*), implying that the transcription rate of protein $p53$ shortens mean FPT or accelerates threshold crossing, whereas its degradation rate increases mean FPT. From figure 5*a,b*, we further observe that for a fixed transcription rate ($g_{p53}$) or degradation rate ($d_{p53}$), mean FPT is a monotonically increasing function of event threshold, implying that event threshold slows down the threshold crossing.

However, the dependence of timing variability on both event threshold and transcription rate (or degradation rate) are shown in figure 5*c,d*. We observe that for a given special event threshold, timing variability monotonically decreases with the increase of transcription rate ($g_{p53}$) (referring to figure 5*c*) while it is a monotonically increasing function of degradation rate ($d_{p53}$) (referring to figure 5*d*), and the minimal timing variability falls on the red line. Figure 5*c* also shows that for a fixed transcription rate ($g_{p53}$), timing variability first decreases and then increases with the increase of event threshold, implying there is an optimal event threshold such that timing variability is least (referring to stair-like line). The stair-like line in figure 5*c* is composed of the points corresponding to the minimal timing

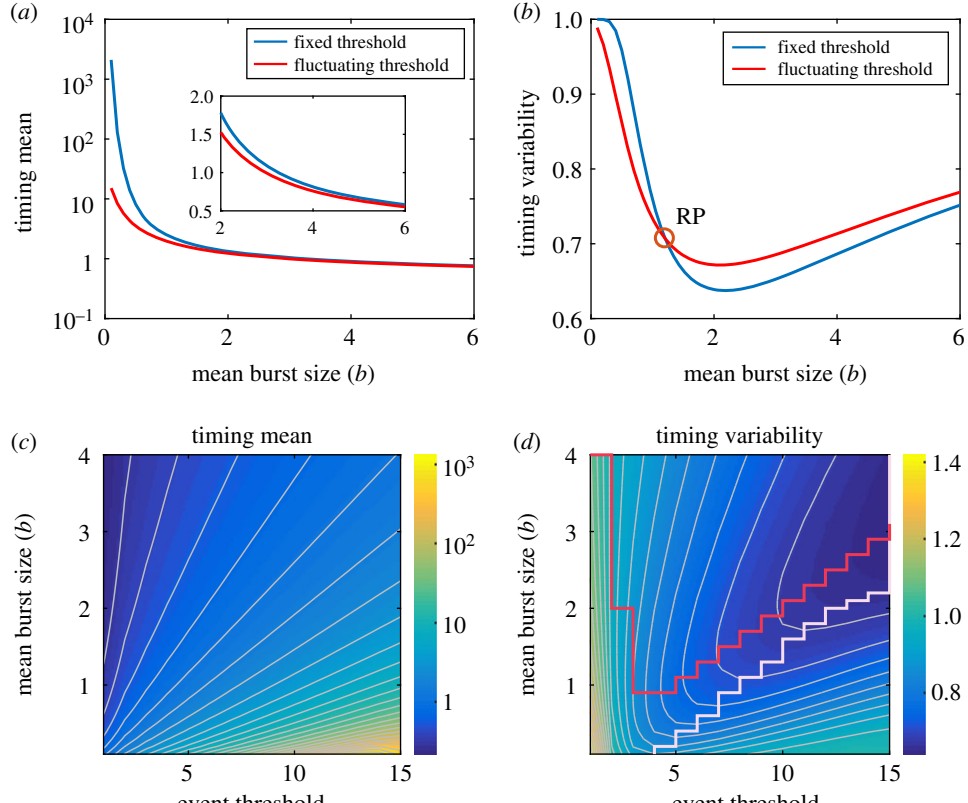

**Figure 6.** Effects of mean burst sizes on event timing. (a,b) Timing mean and timing variability as a function of mean burst size (b) in two cases of fixed (blue curve) and fluctuating (red curve) threshold, where the inset shows a partial enlarged diagram, and RP represents a critical point for reversing. (c,d) Heatmap, respectively, showing timing mean and timing variability as a function of both event threshold and b, where the white curves are contour lines, and the red curve consists of the minimal timing variability in the case of a given event threshold, while the white stair-like line is minimal that in the case of a fixed mean burst size, similar to the star in figure 3c. In (a–d), the parameter values are set as $g_{p53} = 5$, $d_{p53} = 1$, $d_A = 1$ and $p53_{max} = 30$. In (a,b), $b = 0.1 \sim 6$ and $A_{threshold} = g_A = 10$, whereas $b = 0.1 \sim 4$ and $A_{threshold} = g_A = 1 \sim 15$ in (c,d).

variability. From figure 5d, we also observe that for a fixed degradation rate ($d_{p53}$), there exists a minimal timing variability (referring to stair-like line).

In a short, both smaller transcription rates and larger event thresholds or both larger degradation rates and larger event thresholds lead to larger mean FPTs, implying that fewer cancer cells are killed. Moreover, there is minimal timing variability for a fixed transcription or degradation rate, and this result is the same as the result shown in figure 3c. If event threshold is given, then larger transcription rates or smaller degradation rates make timing variability become small, and then enhance the timing precision.

## 3.5. Effect of burst size in p53 on fractional killing of cancer cells

Here, we investigate the influence of burst size on mean FPT and timing variability, with results shown in figure 6. From figure 6a, we observe that mean FPT is a monotonically decreasing function of mean burst size (b) in two cases of fixed and fluctuating threshold, demonstrating mean burst size can shorten mean FPT. Moreover, the red curve of mean FPT in fluctuating threshold case is always below the blue curve in fixed threshold case, implying that fluctuating threshold accelerates threshold crossing. Figure 6b shows the dependence of timing variability on the mean burst size (b) for two different cases of fixed and fluctuating threshold. Timing variability curves firstly decreases and then increases with the increase of mean burst size (b). There is optimal b in figure 6b such that timing variability is least, implying the mean burst size can make timing precision reach the optimality in both fixed and fluctuating threshold cases. Also, there is a critical b (denoted by RP) such that the timing variability in the case of fluctuating threshold is smaller than that in the case of fixed threshold when b is below RP, but the former is larger than the latter if b is beyond RP. It shows that for a small b, threshold fluctuations can reduce the timing variability or raise the precision in the timing.

Further, figure 6*c,d*, respectively, show the heatmap for the dependence of mean FPT and timing variability on both event threshold and mean burst size. From figure 6*c*, we observe that larger mean FPT appear in the region of the right-down corner, which corresponds to both smaller mean burst size and larger event thresholds. Specifically, for a given special event threshold, a larger mean burst size leads to the reduction of mean FPT, implying that translation burst can accelerate response by shortening mean FPT arriving at the fluctuating threshold, which is similar to the results in figure 6*a*. While for a fixed mean burst size, a larger event threshold leads to the increase of mean FPT, implying that fluctuating threshold can slow down response by prolonging mean FPT arriving at the fluctuating threshold. In figure 6*d*, we find that larger timing variability appears approximately in the region of the lower-left corner, which corresponds to both smaller mean burst size and smaller event threshold. Specifically, for a given small event threshold, a smaller mean burst size leads to the increase of timing variability, implying that translation burst can slow down response to the event timing. And there exists an optimal mean burst size such that the timing precision is best for almost large event threshold (referring to red curve in figure 6*d*). In addition, for a fixed large mean burst size, a larger event threshold can reduce timing variability, implying that fluctuating threshold can enhance timing precision. However, there exists an optimal event threshold such that the timing precision is best for almost large mean (referring to white stair-like curve in figure 6*d*).

In short, translation burst (internal noise) accelerates threshold crossing, implying that more cancer cells are killed. There is a critical mean burst size such that translation burst enhances timing precision as the mean burst size is below this critical value, but reduces timing precision as the mean burst size is beyond this critical value. And there is an optimal mean burst size such that the timing precision is best.

Finally, we further discuss another distribution of burst size *B* in electronic supplementary material, S1, since burst size *B* follows a geometric distribution in the above model. We focus on how fluctuations affect mean FPT and timing variability if *B* follows a Poisson distribution. We observe that these results are analogous to those obtained above, implying that burst size distributions have little influence on mean FPT and timing variability, shown in electronic supplementary material, figures S9 and S10. For three other cases of different absorbing domains where burst size follows a Poisson distribution, the mean FPT and timing variability change trends, similar to those in the above respective three cases, as the fluctuating threshold increases. Numerical results are shown in electronic supplementary material, figures S11–S13.

# 4. Conclusion and discussion

While fractional killing is a major impediment to the treatment of cancer, viruses and microbial infections, non-genetic variability plays a pivotal role in fractional killing. Sources of this variability may be complex: apart from molecular noise inherent to gene expression, there is the stochastic fluctuations in apoptotic threshold [2,3]. In this paper, we have systematically investigated a stochastic gene expression system underlying the process of fractional killing, where the cell is killed only when the p53 expression level crosses a fluctuating threshold for the first time. The main contributions and insights can be summarized as follows: (i) fluctuations in apoptotic threshold accelerate timing response, and a faster fluctuation leads to a smaller mean FPT or to killing of more cancer cells; (ii) there is an optimal event threshold such that the timing variability is least; (iii) there is an optimal mean burst size such that the timing repression is best or the timing variability is smallest; (iv) for a high enough threshold, fluctuations in threshold can raise timing precision; and (v) the timescales between transcription and degradation rates can adjust the precision in the timing, independent of the ratio of transcription rate over degradation rate. These results indicate that in contrast to fixed apoptotic thresholds, fluctuating apoptotic threshold can significantly influence the timing of events or killing of cancer cells.

Although we used a simple stochastic model to investigate fractional killing processes involving timing events, our theoretical framework, i.e. a one-dimensional FPT problem with a fluctuating boundary is transformed into a two-dimensional FPT problem with a fixed boundary, can be easily extended to other complex or general cases. In fact, timing events can be attributed to a canonical mechanism of threshold crossing (that can occur in many cellular processes ranging from responses of cells to their environmental cues to cell cycles and circadian clocks), by which a molecular event triggering a cellular behaviour is accumulation to a threshold [17,18,39–42]. In this mechanism, molecules are steadily produced by the cell, and once the molecule number crosses a particular threshold, the behaviour is initiated. Most of these threshold-crossing processes are based on gene

expression, e.g. an activated gene may be required to reach in a precise time a threshold level of gene expression that triggers a specific downstream pathway. However, a gene may reach a critical threshold of expression with substantial cell-to-cell variability even among isogenic cells exposed to the same constant stimulus. This variability is a necessary consequence of the inherently stochastic nature of gene expression [43–49]. Apart from this internal stochastic origin of timing, fluctuating threshold can also result in variability in the event timing required to reach a critical threshold level. It is possible that the intrinsic 'molecular noise' in intracellular processes is responsible for such cell-to-cell variability in the event timing. This is experimentally difficult to verify, but may beg theoretical analysis as done in this paper.

How robust are our results to noise sources and key modelling assumptions? For example, our model only considers the intrinsic noise in gene product levels but ignores the extrinsic noise in gene expression machinery [50,51]. To incorporate such extrinsic noise, one may alter the transcription rate to $k_i Z$ ($k_i$ is an external parameter), where $Z$ may be drawn from an *a priori* probability distribution at the start of gene expression ($t = 0$) and remains fixed till the threshold is reached. Our model also ignored feedback regulation, which, however, exists widely in biological regulatory systems. Recent work has investigated the impact of feedback regulation on the timing of events in the case of fixed threshold [17,18,42]. Interestingly, it was found that there is an optimal feedback strategy to regulate the synthesis of a protein to ensure that an event will occur at a precise time, while minimizing deviations or noise about the mean. Despite this, how feedback regulation controls or impacts the timing of events in the case of fluctuating threshold is unclear. Using our analysis framework, one can also study the effect of feedback regulation on the timing of events in the case of fluctuating threshold. In our case, if changes in burst size, transcription rate or degradation rate are taken as the consequence of feedback regulation, the effect of feedback regulation on the timing of events will become clear in the case of fluctuating threshold. In addition, complex regulatory network composed of apoptosis-related proteins also plays an important role in fractional killing. It is promising for future work to study how cross-talk between the apoptosis pathway and survival pathways affect fractional killing [52].

Regulating some parameter rates of biochemical reactions is an experimental challenge. In our model, the noise regulating threshold crossing has an important impact on apoptosis control strategies. For example, Dar *et al*. had also shown that we can modulate noise in gene expression to enhance threshold crossing without changing mean expression [53]. For this noise control strategy, it will be further studied to regulate both the protein production and degradation rates in our future work. In addition, our model only considers cell apoptotic occurs in the lifetime of the cell, but the cell cycle is not considered in our FPT theoretical framework. Some earlier papers have shown that both cell division and cell cycle arrest could affect a cell fate, and has major implications for anti-cancer therapies. Specially, cell-fate decision happens early in the lifetime of a cell, and the apoptotic fate of daughter cell is to a large extent determined by its mother, and is affected by its mother division [54–56]. If we consider that protein p53 apoptosis occurs before cell division, we can improve our model to discuss their FPT problems by adding to a time constraint on the random variable of equation (2.1), $t \leq t_D$, denoted by $t_D$ the cell cycle, referring to the schematic diagram of electronic supplementary material, figure S14(A) in S1. However, as for the case after cell division, the decision-making on apoptosis is a complex mechanism, referring to electronic supplementary material, figure S14(B) in S1. For example, Chakrabarti *et al*. had shown that sister cell has similar fate and shares the same fate (death or survival) about 80% of the time, regardless of whether they were born before or after cisplatin treatment [56]. When early cell-fate determination phenomenon is also discussed in our model, we should need to record the amount of proteins p53 reaching the cell cycle $t_D$, and then determine the extent that they are away from apoptotic threshold, to obtain the apoptosis probability of daughter cell in the next cycle. This theoretical analysis can provide research ideas and directions for our future research.

Next, we simply discuss potential biological implications of our results in the context of fractional bacterial killing and p53 dynamics.

## 4.1. Connecting theoretical insights to fractional killing

Exposure of an isogenic bacterial population to a cidal antibiotic typically fails to eliminate a small fraction of refractory cells. In order to interpret this phenomenon, Roux *et al*. [5] investigated the basis of fractional cell killing by TRAIL and antibody agonists of DR4 and DR5 receptors. They demonstrated the existence of a threshold in initiator caspase activity (referred to as C8) that must be exceeded for cells to die. Interestingly, they found that, in cells that go on to die, C8 activity rises

rapidly and monotonically until the threshold is reached and mitochondrial outer membrane permeabilization ensues, whereas in cells that survive, C8 activity rises more slowly for 1–4 h, never achieving the level required for death, and then falls back to pre-treatment levels over the next 4–8 h due to proteasome-mediated protein degradation. This finding, which can be reproduced by analysis of our model through the proposed method, implies that *Mycobacterium smegmatis* can dynamically persist in the presence of a drug, and the stable number of cells characterizing this persistence was actually a dynamic state of balanced division and death.

## 4.2. Connecting theoretical insights to drug therapy

Many chemotherapeutic drugs only kill a fraction of cancer cells, limiting their effectiveness. Paek *et al.* [3] used live-cell imaging to check the role of p53 dynamics in fractional killing of colon cancer cells in response to chemotherapy. They found that both surviving and dying cells reach similar levels of p53, indicating that cell death is not determined by a fixed p53 threshold. Instead, a cell's death probability relies on the time and levels of p53. Cells must reach a critical threshold level of p53 to execute apoptosis, and this threshold increases over time. The increase in p53 apoptotic threshold is due to drug-dependent induction of anti-apoptotic genes, predominantly in the inhibitors of apoptosis family. While that study underlined the importance of measuring the dynamics of key players in response to chemotherapy to determine mechanisms of resistance and optimize the timing of combination therapy, our study here provided quantitative results for this importance.

Finally, from a theoretical point of view, our work provides a mathematical and computational framework for studying how fluctuation in threshold influences the statistics of FPT. Our methods can be also extended to the analysis of fluctuations of derivative thresholding [5], integral thresholding [57] and oscillation [58]. Exploring these constraints in more detail will be an important avenue for future research. In addition, analytical results and insights obtained here have broader implications for timing phenomenon in chemical kinetics, epidemic spreading, ecological modelling and statistical physics. Moreover, our methods may allow us to better understand the complex patterns of sequentially ordered biochemical events that are often observed in development and cell-fate decision presumably require an effective control of event timing [59–63].

Ethics. We were not required to complete an ethical assessment prior to conducting our research.

Data accessibility. All code for model implementation is available at the Dryad Digital Repository: https://dx.doi.org/10.5061/dryad.8md722p [64].

Authors' contributions. J.Z. conceived and designed the experiments. B.Q. and J.Z. performed the experiments. B.Q. and J.Z. analysed the data. J.Z. contributed reagents/materials/analysis tools. B.Q., T.Z. and J.Z. wrote the paper.

Competing interests. We have no competing financial interests.

Funding. This work was supported by grant nos. 11931019, 91530320, 11775314, 11475273 and 11631005 from Natural Science Foundation of People's Republic of China; 2014CB964703 from Science and Technology Department, People's Republic of China; 201707010117 from the Science and Technology Program of Guangzhou, People's Republic of China.

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
