## [Reviewer comments · Royal Society Open Science]

Review History

RSOS-190462.R0 (Original submission)

Review form: Reviewer 1

Is the manuscript scientifically sound in its present form?

Yes

Are the interpretations and conclusions justified by the results?

Yes

Is the language acceptable?

No

Is it clear how to access all supporting data?

Not Applicable

Do you have any ethical concerns with this paper?

No

Have you any concerns about statistical analyses in this paper?

No

Recommendation?

Major revision is needed (please make suggestions in comments)

Comments to the Author(s)

This manuscript proposed a stochastic model to simulate dynamics and variation in fractional killing of cancer cells, which is a very important topic in the study of drug resistance. The first-passage time problem was modeled by a stochastic model, and this manuscript derived the analytical distribution of the first-passage time. Interesting results are found for the influence of the fluctuations in nearly every parameter in the model, which may provide insights into the mechanisms of drug resistance. This is an interesting and important work.

The major comment is the writing of this manuscript. Much work is needed to improve the writing. Here I do not give a detailed list since this list may quite long. Please check the manuscript carefully and polish the English writing thoroughly.

The following are some suggestions for the main structure of the manuscript.

- (1) The discussion on page 4 for model introduction may be shortened.
- (2) The subsection title on page 7 actually is the master equation of the model rather than "stochastic model formulation". The stochastic model has already been introduced in Eq (3).
- (3) Subsection from page 10 is too long. (D) "the effectiveness ..." may be the first subsection in Results section. The subsection titles (A, B, C) may be removed.
- (4) There are certain overlaps between the figure legends and main text. It is suggested to write the legends briefly.
- (5) Supplementary Information discussed four cases of different absorbing domains. The main text only discussed case 1 in detail. There are totally 13 supplementary figures that are not mentioned in the manuscript at all. Thus, a subsection (or a paragraph) in Results section may be needed to compare
- (6) In the proposed model, the expression of p53 is modeled by a bursting process but the expression of protein A is Poisson. It is not clear for the results if the expression of protein A is also bursting. Some simulation results may be enough to explain this.

Review form: Reviewer 2

Is the manuscript scientifically sound in its present form?

Yes

Are the interpretations and conclusions justified by the results?

No

Is the language acceptable?

Yes

Is it clear how to access all supporting data?

Yes

Do you have any ethical concerns with this paper?

No

Have you any concerns about statistical analyses in this paper?

No

Recommendation?

Major revision is needed (please make suggestions in comments)

Comments to the Author(s)

In the manuscript "Dynamic variability in apoptotic threshold as a strategy for combating fractional killing", Qiu et al present a theoretical framework based on first passage time (FPT) distributions to study the phenomenon of fractional killing of cells in response to drugs. Building a model for a simple two-gene network of p53 and a gene 'A' that sets a threshold for the p53 level (at which point the cell dies), the authors develop an elegant approach to compute the mean FPT (MFPT) and the variability in FPT for this model. The authors use the Chemical Master Equation to model the two-dimensional state space of p53 and A, and then use the finite state projection method to convert the infinite-dimensional transition probability matrix M into a finite dimensional matrix that is amenable to numerical solutions. The main finding from this theory is that fluctuations in the apoptotic threshold reduce the MFPT compared to a non-fluctuating fixed threshold, implying that fractional killing of cells will be larger in the presence of a fluctuating threshold.

The results presented here are to my mind novel and interesting. The theoretical framework is elegant and provides a powerful approach for understanding the phenomenon of fractional killing of cells in response to drugs. Therefore this work would potentially be of interest to a wide range of researchers, and a valuable addition to the literature.

However, there are a number of places where I feel the manuscript needs to be improved, before it can be published. Some of the most important (and counterintuitive) results have been provided with no deeper analysis of why the results occur in the first place. Limitations of the model in the context of the underlying biology and recent reports in the literature have not been carefully discussed. Finally, a number of statements in the manuscript are not very clear and are potentially misleading, including in my opinion, the title of the manuscript. I elaborate below:

Major points:

(1) The single most important (and interesting) result in this work is the observation that a fluctuating apoptotic threshold leads to shorter MFPT as compared to a non-fluctuating threshold. This is fascinating mainly because it is very counterintuitive to me. Fluctuations in the threshold would result in the threshold decreasing sometimes compared to the average threshold. But at other times, these fluctuations would also result in increasing the threshold over the average, right? Therefore, for some p53 trajectories the FPT will become smaller, but for other trajectories the FPT would increase. Then why should the MFPT decrease in the fluctuating barrier situation? This seems quite counter-intuitive to me, and the authors need to provide an explanation for this observation. Since this is the main result, the authors should provide a detailed discussion with additional calculations/simulations explaining why this result is true.

(2) In various places of the manuscript, the authors refer to the previous paper by Paek et al, Cell 2016 and talk about an apoptotic threshold that increases with time. Fig 1A of this manuscript even provides a graphical example of this situation. However, if I understand correctly, this manuscript deals ONLY with the situation of a threshold that on average is fixed at a constant level - is that correct? The comparison in this work is only between a constant threshold (called 'A_threshold') with no fluctuations vs a threshold that is constant on average but with fluctuations, right? In that case, I would strongly urge the authors to show this situation in Fig 1A instead of showing a threshold that increases with time - since the latter scenario is not being studied in the current manuscript, the current Fig 1A could potentially cause confusion. Related to this point is the use of the terminology "dynamically fluctuating threshold" by the authors in

many places. Why not simply say “fluctuating threshold”? The word “dynamic” tends to be associated with the idea of a threshold that changes on average with time as demonstrated in Paek et al, Cell 2016. The authors should therefore be careful in terms of the terminology they use.

(3) A number of earlier papers have shown that cell-fate decisions (whether the cell will eventually divide or go into cell cycle arrest) seem to happen early in the lifetime of the cell. Therefore the eventual fate outcome of a cell is to a large extent already determined by the cellular state inherited by that cell from its mother at the time of the mother’s division. For example, in the paper “Hidden heterogeneity and circadian-controlled cell fate inferred from single cell lineages”, Nat Comm 2018, the authors looked at cell fates of sister cells and found that they shared the same fate about 80% of the time, regardless of whether they were born before or after cisplatin treatment, indicating early decision making. Conceptually similar results were found using very different techniques in the papers “The proliferation-quiescence decision is controlled by a bifurcation in CDK2 activity at mitotic exit”, Cell 2013 and “Competing memories of mitogen and p53 signalling control cell-cycle entry”, Nature 2017.

If I understand the theory presented here correctly, this phenomenon of early decision making is not captured by the FPT framework, is it? Given that the ‘decision’ of death vs survival in the author’s model happens as a result of reaching an absorbing barrier for the first time, the ‘decision’ making in the authors’ model seems to be happening late in the cell’s lifetime, at the moment the p53 trajectory crosses the level of A. The authors should provide a discussion on this interesting aspect in the context of the papers cited above and their own model. Is this indeed a limitation of their model? If so, can the authors propose some way their FPT model can be improved in future work to account for the early decision-making process in cells?

(4) The title of the paper is in my opinion not a fair representation of the contents of this paper. The authors show that increasing the fluctuations in the absorbing barrier can reduce the MFPT and hence increase the fraction of cells killed. However, in order to achieve this, the production and degradation rates have to be simultaneously increased for the protein A. How can this be achieved realistically, in actual experiments? The authors do not provide any discussion on this important point. Indeed, to my knowledge, this may not be an easy task to achieve. In that case, suggesting that controlling the fluctuations in the threshold level might be a ‘strategy’ to combat fractional killing seems too strong a conclusion in my opinion. The authors should provide a discussion on the feasibility of this ‘strategy’ and also potentially change the title of the manuscript to better reflect their interesting, but theoretical findings.

Minor Points:

(1) In many places throughout the manuscript, the authors use the terminology “fast fluctuations” and “slow fluctuations”. Would it be more accurate to say “large fluctuations” and “small fluctuations”? By increasing the magnitude of the production and degradation rates of p53 or A, the amplitude of the noise increases, not how fast the fluctuations occur.

(2) On page 6 of the manuscript, the line “Note that the more threshold-crossing events are, the fewer cancer cells are killed....” is quite confusing. Did the authors mean “more” cancer cells are killed?

(3) On page 22, the authors say “However, the results in the case of timing variability is almost converse to those...”. This statement seems incorrect to me after looking at Fig 5 - it seems to me that BOTH the MFPT and the timing variability are decreasing with increase of the p53 transcription rate.

(4) In the heatmaps shown, for example in Figures 5 and 6, the authors may contemplate adding contour lines to aid the eye. It’s a little difficult to see the trends in the heatmaps as shown in the current figures.

Decision letter (RSOS-190462.R0)

02-Oct-2019

Dear Dr Zhang,

The editors assigned to your paper ("Dynamic variability in apoptotic threshold as a strategy for combating fractional killing") have now received comments from reviewers. We would like you to revise your paper in accordance with the referee and Associate Editor suggestions which can be found below (not including confidential reports to the Editor). Please note this decision does not guarantee eventual acceptance.

Please submit a copy of your revised paper before 25-Oct-2019. Please note that the revision deadline will expire at 00.00am on this date. If we do not hear from you within this time then it will be assumed that the paper has been withdrawn. In exceptional circumstances, extensions may be possible if agreed with the Editorial Office in advance. We do not allow multiple rounds of revision so we urge you to make every effort to fully address all of the comments at this stage. If deemed necessary by the Editors, your manuscript will be sent back to one or more of the original reviewers for assessment. If the original reviewers are not available, we may invite new reviewers.

- Data accessibility

<http://datadryad.org/submit?journalID=RSOS&manu=RSOS-190462>

- Competing interests

- Authors' contributions

- Acknowledgements

- Funding statement

Kind regards,

Lianne Parkhouse

on behalf of Dr Anna Marciniak-Czochra (Associate Editor) and Mark Chaplain (Subject Editor)
openscience@royalsociety.org

Reviewers' Comments to Author:

Reviewer: 1

Comments to the Author(s)

This manuscript proposed a stochastic model to simulate dynamics and variation in fractional killing of cancer cells, which is a very important topic in the study of drug resistance. The first-passage time problem was modeled by a stochastic model, and this manuscript derived the analytical distribution of the first-passage time. Interesting results are found for the influence of the fluctuations in nearly every parameter in the model, which may provide insights into the mechanisms of drug resistance. This is an interesting and important work.

The major comment is the writing of this manuscript. Much work is needed to improve the writing. Here I do not give a detailed list since this list may quite long. Please check the manuscript carefully and polish the English writing thoroughly.

The following are some suggestions for the main structure of the manuscript.

- (1) The discussion on page 4 for model introduction may be shortened.
- (2) The subsection title on page 7 actually is the master equation of the model rather than “stochastic model formulation”. The stochastic model has already been introduced in Eq (3).
- (3) Subsection from page 10 is too long. (D) “the effectiveness ...” may be the first subsection in Results section. The subsection titles (A, B, C) may be removed.
- (4) There are certain overlaps between the figure legends and main text. It is suggested to write the legends briefly.
- (5) Supplementary Information discussed four cases of different absorbing domains. The main text only discussed case 1 in detail. There are totally 13 supplementary figures that are not mentioned in the manuscript at all. Thus, a subsection (or a paragraph) in Results section may be needed to compare
- (6) In the proposed model, the expression of p53 is modeled by a bursting process but the expression of protein A is Poisson. It is not clear for the results if the expression of protein A is also bursting. Some simulation results may be enough to explain this.

Reviewer: 2

Comments to the Author(s)

In the manuscript “Dynamic variability in apoptotic threshold as a strategy for combating fractional killing”, Qiu et al present a theoretical framework based on first passage time (FPT) distributions to study the phenomenon of fractional killing of cells in response to drugs. Building a model for a simple two-gene network of p53 and a gene ‘A’ that sets a threshold for the p53 level (at which point the cell dies), the authors develop an elegant approach to compute the mean FPT (MFPT) and the variability in FPT for this model. The authors use the Chemical Master Equation to model the two-dimensional state space of p53 and A, and then use the finite state projection method to convert the infinite-dimensional transition probability matrix M into a finite dimensional matrix that is amenable to numerical solutions. The main finding from this theory is that fluctuations in the apoptotic threshold reduce the MFPT compared to a non-fluctuating fixed threshold, implying that fractional killing of cells will be larger in the presence of a fluctuating threshold.

The results presented here are to my mind novel and interesting. The theoretical framework is elegant and provides a powerful approach for understanding the phenomenon of fractional killing of cells in response to drugs. Therefore this work would potentially be of interest to a wide range of researchers, and a valuable addition to the literature.

However, there are a number of places where I feel the manuscript needs to be improved, before it can be published. Some of the most important (and counterintuitive) results have been provided with no deeper analysis of why the results occur in the first place. Limitations of the model in the context of the underlying biology and recent reports in the literature have not been carefully discussed. Finally, a number of statements in the manuscript are not very clear and are potentially misleading, including in my opinion, the title of the manuscript. I elaborate below:

Major points:

(1) The single most important (and interesting) result in this work is the observation that a fluctuating apoptotic threshold leads to shorter MFPT as compared to a non-fluctuating threshold. This is fascinating mainly because it is very counterintuitive to me. Fluctuations in the threshold would result in the threshold decreasing sometimes compared to the average threshold. But at other times, these fluctuations would also result in increasing the threshold over the average, right? Therefore, for some p53 trajectories the FPT will become smaller, but for other trajectories the FPT would increase. Then why should the MFPT decrease in the fluctuating barrier situation? This seems quite counter-intuitive to me, and the authors need to provide an explanation for this observation. Since this is the main result, the authors should provide a detailed discussion with additional calculations/simulations explaining why this result is true.

(2) In various places of the manuscript, the authors refer to the previous paper by Paek et al, Cell 2016 and talk about an apoptotic threshold that increases with time. Fig 1A of this manuscript even provides a graphical example of this situation. However, if I understand correctly, this manuscript deals ONLY with the situation of a threshold that on average is fixed at a constant level - is that correct? The comparison in this work is only between a constant threshold (called 'A_threshold') with no fluctuations vs a threshold that is constant on average but with fluctuations, right? In that case, I would strongly urge the authors to show this situation in Fig 1A instead of showing a threshold that increases with time - since the latter scenario is not being studied in the current manuscript, the current Fig 1A could potentially cause confusion. Related to this point is the use of the terminology "dynamically fluctuating threshold" by the authors in many places. Why not simply say "fluctuating threshold"? The word "dynamic" tends to be associated with the idea of a threshold that changes on average with time as demonstrated in Paek et al, Cell 2016. The authors should therefore be careful in terms of the terminology they use.

(3) A number of earlier papers have shown that cell-fate decisions (whether the cell will eventually divide or go into cell cycle arrest) seem to happen early in the lifetime of the cell. Therefore the eventual fate outcome of a cell is to a large extent already determined by the cellular state inherited by that cell from its mother at the time of the mother's division. For example, in the paper "Hidden heterogeneity and circadian-controlled cell fate inferred from single cell lineages", Nat Comm 2018, the authors looked at cell fates of sister cells and found that they shared the same fate about 80% of the time, regardless of whether they were born before or after cisplatin treatment, indicating early decision making. Conceptually similar results were found using very different techniques in the papers "The proliferation-quiescence decision is controlled by a bifurcation in CDK2 activity at mitotic exit", Cell 2013 and "Competing memories of mitogen and p53 signalling control cell-cycle entry", Nature 2017.

If I understand the theory presented here correctly, this phenomenon of early decision making is not captured by the FPT framework, is it? Given that the 'decision' of death vs survival in the author's model happens as a result of reaching an absorbing barrier for the first time, the 'decision' making in the authors' model seems to be happening late in the cell's lifetime, at the moment the p53 trajectory crosses the level of A. The authors should provide a discussion on this interesting aspect in the context of the papers cited above and their own model. Is this indeed a limitation of their model? If so, can the authors propose some way their FPT model can be improved in future work to account for the early decision-making process in cells?

(4) The title of the paper is in my opinion not a fair representation of the contents of this paper. The authors show that increasing the fluctuations in the absorbing barrier can reduce the MFPT and hence increase the fraction of cells killed. However, in order to achieve this, the production and degradation rates have to be simultaneously increased for the protein A. How can this be achieved realistically, in actual experiments? The authors do not provide any discussion on this important point. Indeed, to my knowledge, this may not be an easy task to achieve. In that case, suggesting that controlling the fluctuations in the threshold level might be a 'strategy' to combat

fractional killing seems too strong a conclusion in my opinion. The authors should provide a discussion on the feasibility of this 'strategy' and also potentially change the title of the manuscript to better reflect their interesting, but theoretical findings.

Minor Points:

- (1) In many places throughout the manuscript, the authors use the terminology "fast fluctuations" and "slow fluctuations". Would it be more accurate to say "large fluctuations" and "small fluctuations"? By increasing the magnitude of the production and degradation rates of p53 or A, the amplitude of the noise increases, not how fast the fluctuations occur.
- (2) On page 6 of the manuscript, the line "Note that the more threshold-crossing events are, the fewer cancer cells are killed...." is quite confusing. Did the authors mean "more" cancer cells are killed?
- (3) On page 22, the authors say "However, the results in the case of timing variability is almost converse to those...". This statement seems incorrect to me after looking at Fig 5 - it seems to me that BOTH the MFPT and the timing variability are decreasing with increase of the p53 transcription rate.
- (4) In the heatmaps shown, for example in Figures 5 and 6, the authors may contemplate adding contour lines to aid the eye. It's a little difficult to see the trends in the heatmaps as shown in the current figures.

Author's Response to Decision Letter for (RSOS-190462.R0)

See Appendix A.

RSOS-190462.R1 (Revision)

Review form: Reviewer 1

Is the manuscript scientifically sound in its present form?

Yes

Are the interpretations and conclusions justified by the results?

Yes

Is the language acceptable?

Yes

Do you have any ethical concerns with this paper?

No

Have you any concerns about statistical analyses in this paper?

No

Recommendation?

Accept as is

Comments to the Author(s)

The authors have addressed the comments in the first report very well. The quality of this paper was improved substantially, in both research and writing. There is no further comments in this second report.

Review form: Reviewer 2

Is the manuscript scientifically sound in its present form?

Yes

Are the interpretations and conclusions justified by the results?

Yes

Is the language acceptable?

Yes

Do you have any ethical concerns with this paper?

No

Have you any concerns about statistical analyses in this paper?

No

Recommendation?

Accept as is

Comments to the Author(s)

The authors have now satisfactorily responded to all points and have updated their manuscript accordingly. I thank the authors for changing the title of the manuscript to a more appropriate one. Overall I think this is a nice piece of work, and is now ready for publishing.

Decision letter (RSOS-190462.R1)

20-Jan-2020

Dear Dr Zhang,

It is a pleasure to accept your manuscript entitled "Stochastic fluctuations in apoptotic threshold of tumor cells can enhance apoptosis and combat fractional killing" in its current form for publication in Royal Society Open Science. The comments of the reviewer(s) who reviewed your manuscript are included at the foot of this letter.

Best regards,

on behalf of the Associate Editor, and Professor Mark Chaplain (Subject Editor)
openscience@royalsociety.org

Reviewer comments to Author:

Reviewer: 1
Comments to the Author(s)

The authors have addressed the comments in the first report very well. The quality of this paper was improved substantially, in both research and writing. There is no further comments in this second report.

Reviewer: 2
Comments to the Author(s)

The authors have now satisfactorily responded to all points and have updated their manuscript accordingly. I thank the authors for changing the title of the manuscript to a more appropriate one. Overall I think this is a nice piece of work, and is now ready for publishing.

Appendix A

Response to reviewer #1

Thanks for your comments and suggestions. According to your review report, we have made revisions. Details of response to your moments are as follow.

This manuscript proposed a stochastic model to simulate dynamics and variation in fractional killing of cancer cells, which is a very important topic in the study of drug resistance. The first-passage time problem was modeled by a stochastic model, and this manuscript derived the analytical distribution of the first-passage time. Interesting results are found for the influence of the fluctuations in nearly every parameter in the model, which may provide insights into the mechanisms of drug resistance. This is an interesting and important work.

The major comment is the writing of this manuscript. Much work is needed to improve the writing. Here I do not give a detailed list since this list may quite long. Please check the manuscript carefully and polish the English writing thoroughly.

Response: We appreciate your positive feedback. We have made some revision and improved clarity in the current manuscript.

Questions:

The following are some suggestions for the main structure of the manuscript.

(1) The discussion on page 4 for model introduction may be shortened.

Response: Thank you for raising this point. This discussion on page 4 has been shortened.

(2) The subsection title on page 7 actually is the master equation of the model rather than "stochastic model formulation". The stochastic model has already been introduced in Eq (3).

Response: Thank you for the great advice. We have fixed the typos in this subsection title, and this title is rewritten to " Master equation for FPT problem with a fluctuating threshold".

(3) Subsection from page 10 is too long. (D) "the effectiveness ..." may be the first subsection in Results section. The subsection titles (A, B, C) may be removed.

Response: Thank you for giving constructive advice. We have removed these subsection titles and adjusted the effectiveness of the truncation algorithm into the Results section.

(4) There are certain overlaps between the figure legends and main text. It is suggested to write the legends briefly.

Response: Thank you for carefully reading our manuscript and pointing out this point. We have addressed the legend of all figures, and rewritten briefly it.

(5) Supplementary Information discussed four cases of different absorbing domains. The main text

only discussed case 1 in detail. There are totally 13 supplementary figures that are not mentioned in the manuscript at all. Thus, a subsection (or a paragraph) in Results section may be needed to compare.

Response: Thanks again for raising great suggestion. We have added a paragraph to discuss four different cases in Results section based on the *SI Supplementary Information*. In the *SI supplementary information*, two main parts are discussed. One is to expand the absorbing domain of main file and discuss respectively the timing mean (i.e., mean FPT) and timing variability in the other three cases of different absorbing domains. And, in the subsection "*Stochastic fluctuations in apoptotic thresholds can accelerate tumor cell apoptosis*", we have added a paragraph to discuss their numerical results and difference between four different cases. The other is to consider different burst size distribution of protein P53, assuming here that the burst size B follows a Poisson distribution, and discuss its influence on the timing mean (i.e., mean FPT) and timing variability. For their numerical results and difference between four different cases, we also have added in the last paragraph of the subsection "*Effect of burst size in p53 on fractional killing of cancer cells*".

(6) In the proposed model, the expression of p53 is modeled by a bursting process but the expression of protein A is Poisson. It is not clear for the results if the expression of protein A is also bursting. Some simulation results may be enough to explain this.

Response: Thank you for raising this point. Our model does not discuss its burst size in the expression of protein A. Note that the threshold crossing is a complex biochemical process and the threshold are not always a protein level. Our model uses the birth-death process of protein to mimic the fluctuations of threshold. Indeed, the threshold fluctuations can be also modeled as a bursting process. This will be investigated in further work.

Response to reviewer #2

Thanks for your comments and suggestions. According to your review report, we have made revisions. Details of response to your moments are as follow.

In the manuscript “Dynamic variability in apoptotic threshold as a strategy for combating fractional killing”, Qiu et al present a theoretical framework based on first passage time (FPT) distributions to study the phenomenon of fractional killing of cells in response to drugs. Building a model for a simple two-gene network of p53 and a gene ‘A’ that sets a threshold for the p53 level (at which point the cell dies), the authors develop an elegant approach to compute the mean FPT (MFPT) and the variability in FPT for this model. The authors use the Chemical Master Equation to model the two-dimensional state space of p53 and A, and then use the finite state projection method to convert the infinite-dimensional transition probability matrix M into a finite dimensional matrix that is amenable to numerical solutions. The main finding from this theory is that fluctuations in the apoptotic threshold reduce the MFPT compared to a non-fluctuating fixed threshold, implying that fractional killing of cells will be larger in the presence of a fluctuating threshold.

The results presented here are to my mind novel and interesting. The theoretical framework is elegant and provides a powerful approach for understanding the phenomenon of fractional killing of cells in response to drugs. Therefore this work would potentially be of interest to a wide range of researchers, and a valuable addition to the literature.

However, there are a number of places where I feel the manuscript needs to be improved, before it can be published. Some of the most important (and counterintuitive) results have been provided with no deeper analysis of why the results occur in the first place. Limitations of the model in the context of the underlying biology and recent reports in the literature have not been carefully discussed. Finally, a number of statements in the manuscript are not very clear and are potentially misleading, including in my opinion, the title of the manuscript. I elaborate below:

Response: Thank you for your positive feedback. We have improved clarity in the revised manuscript. Some supplements and discussions have also been made on the issues you mentioned.

Major points:

(1) The single most important (and interesting) result in this work is the observation that a fluctuating apoptotic threshold leads to shorter MFPT as compared to a non-fluctuating threshold. This is fascinating mainly because it is very counterintuitive to me. Fluctuations in the threshold would result in the threshold decreasing sometimes compared to the average threshold. But at other times, these fluctuations would also result in increasing the threshold over the average, right? Therefore, for some p53 trajectories the FPT will become smaller, but for other trajectories the FPT would increase. Then why should the MFPT decrease in the fluctuating barrier situation? This seems quite counter-intuitive to me, and the authors need to provide an explanation for this observation. Since this is the main result, the authors should provide a detailed discussion with additional calculations/simulations explaining why this result is true.

Response: Thank you for the great advice. You are right for the understanding of fluctuations in threshold. We also are very surprised at this counterintuitive result that the MFPT decrease in the fluctuating barrier situation, but a large number of numerical simulations confirms this result. Moreover, we can explain this result theoretically. And, we have added to a paragraph to explain this result in the subsection “Stochastic fluctuations in apoptotic thresholds can accelerate tumor cell apoptosis”, that is,

“Note that our above results can also be explained theoretically (0,0). For the case of fixed threshold, we define $T_{\text{fixed}} = \min\{t : p53(t) \geq \langle A(t) \rangle | p53(0) = p53_0, A(0) = A_0\}$ is the apoptotic timing of random variable T_{fixed} , where $\langle A(t) \rangle$ is mean apoptotic threshold. Obviously, it is different with the defined variable T of Eq. (1) in the case of fluctuating threshold, rewritten by $T_{\text{fluctuating}} = \min\{t : p53(t) \geq A(t) | p53(0) = p53_0, A(0) = A_0\}$. As for nonlinear function, these two mean FPTs is not equal, i.e., $\langle T_{\text{fluctuating}}(A(t)) \rangle \neq \langle T_{\text{fixed}}(\langle A(t) \rangle) \rangle$, implying that the average apoptotic timing response probability is generally not the same as the timing response probability for the average apoptotic threshold signal. Although it is difficult to deduce their size relationship directly theoretically in our model, according to numerical calculation results, we can show this result that the mean FPF in the case of fluctuation threshold is less than that in the case of fixed threshold.”

References:

(36) Paulsson J, G. Berg O, Ehrenberg M. 2000. Stochastic focusing: Fluctuation-enhanced sensitivity of intracellular regulation. Proc. Natl. Acad. Sci. USA. 97: 7148-7153. (<https://doi.org/10.1073/pnas.110057697>)

(37)Samoilov MS, Arkin AP. 2006. Deviant effects in molecular reaction pathways. Nat. Biotechnology. 24, 1235-1240. (<https://doi.org/10.1038/nbt1253>)

(2) *In various places of the manuscript, the authors refer to the previous paper by Paek et al, Cell 2016 and talk about an apoptotic threshold that increases with time. Fig 1A of this manuscript even provides a graphical example of this situation. However, if I understand correctly, this manuscript deals ONLY with the situation of a threshold that on average is fixed at a constant level – is that correct? The comparison in this work is only between a constant threshold (called ‘A_threshold’) with no fluctuations vs a threshold that is constant on average but with fluctuations, right? In that case, I would strongly urge the authors to show this situation in Fig 1A instead of showing a threshold that increases with time – since the latter scenario is not being studied in the current manuscript, the current Fig 1A could potentially cause confusion. Related to this point is the use of the terminology “dynamically fluctuating threshold” by the authors in many places. Why not simply say “fluctuating threshold”? The word “dynamic” tends to be associated with the idea of a threshold that changes on average with time as demonstrated in Paek et al, Cell 2016. The authors should therefore be careful in terms of the*

terminology they use.

Response: Thanks again for detailed reading our paper. Your understanding is correct. We only discuss that the fluctuations in apoptotic threshold varies with time, but we did not investigate the monotonous increase of this threshold with respect to time. Thus, we have modified *Fig 1A* and deleted the confusing word " dynamically " or "dynamic" and some typos to improve the accuracy of the terminology of our paper.

(3) *A number of earlier papers have shown that cell-fate decisions (whether the cell will eventually divide or go into cell cycle arrest) seem to happen early in the lifetime of the cell. Therefore the eventual fate outcome of a cell is to a large extent already determined by the cellular state inherited by that cell from its mother at the time of the mother's division. For example, in the paper "Hidden heterogeneity and circadian-controlled cell fate inferred from single cell lineages", Nat Comm 2018, the authors looked at cell fates of sister cells and found that they shared the same fate about 80% of the time, regardless of whether they were born before or after cisplatin treatment, indicating early decision making. Conceptually similar results were found using very different techniques in the papers "The proliferation-quiescence decision is controlled by a bifurcation in CDK2 activity at mitotic exit", Cell 2013 and "Competing memories of mitogen and p53 signalling control cell-cycle entry", Nature 2017.*

If I understand the theory presented here correctly, this phenomenon of early decision making is not captured by the FPT framework, is it? Given that the 'decision' of death vs survival in the author's model happens as a result of reaching an absorbing barrier for the first time, the 'decision' making in the authors' model seems to be happening late in the cell's lifetime, at the moment the p53 trajectory crosses the level of A. The authors should provide a discussion on this interesting aspect in the context of the papers cited above and their own model. Is this indeed a limitation of their model? If so, can the authors propose some way their FPT model can be improved in future work to account for the early decision-making process in cells?

Response: Thank you for his constructive advice. Your understanding is correct for FPT framework. Our model does not discuss the phenomenon of early decision making. If the cell-cycle is also introduced into our FPT framework, then we need to analyze whether the cell apoptosis occurs before or after the cell division, and we add their schematic diagrams shown in Fig. S14 in S1 Supporting information in revised manuscript. This interesting question will be discussed in future research. In our discussion, we also have added to a paragraph to discuss FPT question involved the early cell-fate decision . That is,

" In addition, our model only consider cell apoptotic occurs in the lifetime of the cell, but the cell cycle is not considered in our FPT theoretical framework. Some earlier papers have shown that both cell division and cell cycle arrest could affect a cell fate, and has major implications for anti-cancer therapies. Specially, cell-fate decision happens early in the lifetime of a cell, and the apoptotic fate of daughter cell is to a large extent determined by its mother, and is affected by its mother division (错误!未找到引用源。 -错误!未找到引用源。). If we consider that protein p53 apoptosis

occurs before cell division, we can improve our model to discuss their FPT problems by adding to a time constraint on the random variable of Eq. (1), $t \leq t_D$, denoted by t_D the cell cycle, referring to the schematic diagram of Fig. S14(A) in S1 Supporting information. However, as for the case after cell division, the decision-making on apoptosis is a complex mechanism, referring to Fig. S14(B) in S1 Supporting information. For example, Chakrabarti R, *et al.* had shown that sister cell has similar fate and shares the same fate (death or survival) about 80% of the time, regardless of whether they were born before or after cisplatin treatment (错误!未找到引用源。). When early cell-fate determination phenomenon is also discussed in our model, we should need to record the amount of proteins p53 reaching the cell cycle t_D , and then determine the extent that they are away from apoptotic threshold, to obtain the apoptosis probability of daughter cell in the next cycle. This theoretical analysis can provide research ideas and directions for our future research. "

(4) *The title of the paper is in my opinion not a fair representation of the contents of this paper. The authors show that increasing the fluctuations in the absorbing barrier can reduce the MFPT and hence increase the fraction of cells killed. However, in order to achieve this, the production and degradation rates have to be simultaneously increased for the protein A. How can this be achieved realistically, in actual experiments? The authors do not provide any discussion on this important point. Indeed, to my knowledge, this may not be an easy task to achieve. In that case, suggesting that controlling the fluctuations in the threshold level might be a 'strategy' to combat fractional killing seems too strong a conclusion in my opinion. The authors should provide a discussion on the feasibility of this 'strategy' and also potentially change the title of the manuscript to better reflect their interesting, but theoretical findings.*

Response: Thanks again for your constructive advice. Yes, it is a challenge to control noise by mediate the production and degradation rates simultaneously. However, in some instances we can achieve the goal. For example, Dar and Hosmane *et al.* show that modulating noise in gene expression without changing mean expression to enhance threshold crossing is feasible. We have added some explanation in the discussion section as follows:

"Regulating some parameters rate of biochemical reactions is an experimental challenge. In our model, the noise regulating threshold crossing has an important impact on apoptosis control strategies. For example, Dar RD, *et al.* had also shown that we can modulate noise in gene expression to enhance threshold crossing without changing mean expression (0). For this noise control strategy, it will be further studied to regulate both the protein production and degradation rates in our future work."

In addition, we have changed the title of our manuscript. For this noise control strategy, combined with some experimental data and results, it will be studied in our future research work.

Reference:

(53) Dar RD, Hosmane NN, Arkin MR, Siliciano RF, Weinberger LS. 2014. Screening for noise in gene expression identifies drug synergies. *Science*. 344: 1392-1396. (<https://doi.org/10.1126/science.1250220>)

Minor Points:

(1) *In many places throughout the manuscript, the authors use the terminology “fast fluctuations” and “slow fluctuations”. Would it be more accurate to say “large fluctuations” and “small fluctuations”? By increasing the magnitude of the production and degradation rates of p53 or A, the amplitude of the noise increases, not how fast the fluctuations occur.*

Response: Thank you for raising this point. We explain the relationship between “fast (or slow) fluctuations” and “large (or small) fluctuations” as follows: " Small α_{p53} or α_A corresponds to slow fluctuations whereas large α_{p53} or α_A to fast fluctuations " in the subsection " *Fast fluctuations in apoptotic threshold can lead to killing of more cancer cells*". Moreover, the “fast (or slow) fluctuations” is mainly used by regulating the fluctuation frequency. Thus, our model uses the terminology “fast fluctuations” and “slow fluctuations”.

(2) *On page 6 of the manuscript, the line “Note that the more threshold-crossing events are, the fewer cancer cells are killed....” is quite confusing. Did the authors mean “more” cancer cells are killed?*

Response: Thank you for pointing out the typos. We have changed the sentence to " Note that the more threshold-crossing events are, the more cancer cells are killed ".

(3) *On page 22, the authors say “However, the results in the case of timing variability is almost converse to those...”. This statement seems incorrect to me after looking at Fig 5 – it seems to me that BOTH the MFPT and the timing variability are decreasing with increase of the p53 transcription rate.*

Response: Thank you for carefully reading our manuscript and pointing out the mistakes. This question has been addressed, and this result is described in revised manuscript as follow: " Moreover, there is minimal timing variability for a fixed transcription or degradation rate, and this result is same as the result shown in Fig. 3C. If event threshold is given, then larger transcription rates or smaller degradation rates make timing variability become small, and then enhance the timing precision."

(4) *In the heatmaps shown, for example in Figures 5 and 6, the authors may contemplate adding contour lines to aid the eye. It’s a little difficult to see the trends in the heatmaps as shown in the current figures.*

Response: Thank again the reviewer for raising great suggestion. We have added to contour lines in the Figs. 5 and 6.